## REPORT

# Lysosome damage triggers direct ATG8 conjugation and ATG2 engagement via non-canonical autophagy

Jake Cross[1], Joanne Durgan[1], David G. McEwan[2], Matthew Tayler[1], Kevin M. Ryan[2], and Oliver Florey[1]

**Cells harness multiple pathways to maintain lysosome integrity, a central homeostatic process. Damaged lysosomes can be repaired or targeted for degradation by lysophagy, a selective autophagy process involving ATG8/LC3. Here, we describe a parallel ATG8/LC3 response to lysosome damage, mechanistically distinct from lysophagy. Using a comprehensive series of biochemical, pharmacological, and genetic approaches, we show that lysosome damage induces non-canonical autophagy and Conjugation of ATG8s to Single Membranes (CASM). Following damage, ATG8s are rapidly and directly conjugated onto lysosome membranes, independently of ATG13/WIPI2, lipidating to PS (and PE), a molecular hallmark of CASM. Lysosome damage drives V-ATPase V0-V1 association, direct recruitment of ATG16L1 via its WD40-domain/K490A, and is sensitive to *Salmonella* SopF. Lysosome damage-induced CASM is associated with formation of dynamic, LC3A-positive tubules, and promotes robust LC3A engagement with ATG2, a lipid transfer protein central to lysosome repair. Together, our data identify direct ATG8 conjugation as a rapid response to lysosome damage, with important links to lipid transfer and dynamics.**

## Introduction

The lysosome is an acidic, catabolic organelle that plays a central role in the degradation of extra- and intra-cellular cargos (Ballabio and Bonifacino, 2020) to maintain cellular homeostasis and health. Lysosomes also play a vital signaling function, acting as hubs to sense, integrate, and respond to changes in nutrient status, metabolic signals, and cellular stress (Perera and Zoncu, 2016). Given these fundamental functions, the maintenance of lysosomal integrity is of paramount importance for cellular and organismal homeostasis. Many agents and conditions can threaten this integrity by damaging late endosomal or lysosomal compartments through partial lysosomal membrane permeabilization (LMP) or full rupture of lysosomes. Damage can be caused by endocytosed materials, such as silica or urate crystals, that mechanically rupture the membrane, as well as neurotoxic aggregates, pathogens, and membrane lipid changes associated with aging (Wang et al., 2018; Zhen et al., 2021). To defend against such perturbations, cells have developed sophisticated defense mechanisms that together comprise the lysosomal damage response (LDR).

Small-scale lysosome damage can be repaired via an ESCRT-mediated membrane remodeling process that is dependent on Ca²⁺ release (Radulovic et al., 2018; Scheffer et al., 2014; Skowyra et al., 2018). Additionally, a parallel repair pathway, termed phosphoinositide-initiated membrane tethering and lipid transport (PITT), has recently been described (Tan and Finkel, 2022). During PITT, PI4K2A is recruited to damaged lysosomes to increase the level of PI4P. This in turn engages ORP-VAPA lipid transfer complexes to form ER contact sites which transfer phosphatidylserine (PS) and cholesterol to the lysosome membrane (Radulovic et al., 2022). Subsequently, the lipid transfer protein, ATG2, is recruited, supplying further lipids to repair the damaged lysosome. Another protein, LRRK2, is also recruited to stressed lysosomes, where it plays an important role in restoring integrity (Eguchi et al., 2018; Herbst et al., 2020), through the generation of lysosome tubules, in a process termed LYsosomal Tubulation/sorting driven by LRRK2 (LYTL; Bonet-Ponce et al., 2020). These repair mechanisms act in parallel to ensure robust protection of the lysosomal system.

In the case of more profound membrane damage, lysosomes can instead be cleared by a selective autophagy process called lysophagy (Maejima et al., 2013). Lysophagy involves the recognition of luminal galectins, which, when exposed to the cytosol, promote the recruitment of the ULK/ATG13/FIP200 autophagy initiation complex. This complex acts in concert with Vps34 and PI3P generation to promote the de novo formation of a double-membrane autophagosome that sequesters the damaged lysosome (Maejima et al., 2013). A key feature of this autophagosome formation is the involvement of a ubiquitin-like conjugation system, including ATG16L1, to target the conjugation of ATG8 proteins (LC3A/B/C and GABARAP/L1/L2 families) to phosphatidylethanolamine (PE) in the forming autophagosome membrane (Fujita et al., 2013; Ichimura et al., 2000). ATG8

---

¹Signalling Programme, Babraham Institute, Cambridge, UK;   ²Tumour Cell Death and Autophagy Laboratory, Cancer Research UK Beatson Institute, Glasgow, UK.

Correspondence to Oliver Florey: oliver.florey@babraham.ac.uk.

proteins act with WIPI4 to recruit ATG2, initiating contacts with the ER and transferring lipids to support autophagosome expansion (Bozic et al., 2020; Osawa et al., 2019; Valverde et al., 2019). ATG2 thus functions in both the PITT pathway and during classical lysophagy as a central player in lysosome homeostasis.

A subset of autophagy proteins can also function in a parallel "non-canonical autophagy" pathway, defined by the conjugation of ATG8 to single membranes of the endolysosomal system (CASM; Durgan and Florey, 2022). CASM is implicated in a wide range of processes, including phagocytosis, endocytosis, and response to infection (viral, bacterial, yeast), with important functions in immunity (Heckmann et al., 2017) and neurodegeneration (Heckmann et al., 2019, 2020). Mechanistically, CASM is independent of canonical autophagy and does not require the ULK/ATG13/FIP200 initiation complex (Florey et al., 2015; Jacquin et al., 2017; Martinez et al., 2015). Instead, CASM engages the ATG16L1 complex to promote ATG8 lipidation directly onto single, endolysosomal membranes via conjugation to PE and also to phosphatidylserine (PS); ATG8-PS conjugation is a specific molecular signature of CASM (Durgan et al., 2021). A broad set of stimuli can activate CASM, including ROS generation, pathogens, lysosomotropic drugs, ionophores, activation of the STING pathway, and agonists of the lysosomal $Ca^{2+}$ channel TRPML1. These diverse stimuli all share the common feature of perturbing the ionic and pH balance of the endolysosomal system (Durgan and Florey, 2022). In response to elevated lysosomal pH, the V1 and V0 sectors of V-ATPase engage at the lysosome membrane to drive reacidification. This enhanced engagement also directly recruits ATG16L1, specifying the site for ATG8 lipidation by CASM. This V-ATPase-ATG16L1 (VAIL) axis is dependent on critical residues in the ATG16L1 C-terminal WD40 domain, including K490 (Fletcher et al., 2018), and can be inhibited by the *Salmonella* effector protein, SopF (Fischer et al., 2020; Hooper et al., 2022; Ulferts et al., 2021; Xu et al., 2019). While the functional significance of the CASM pathway is now well established, the precise downstream mechanisms remain to be fully defined. Recent work has defined a CASM pathway involved in lysosome biogenesis, and this field lies open for further investigation.

Interestingly, some recent studies have suggested that an unconventional form of ATG8 lipidation may occur at damaged lysosomes (Jia et al., 2022; Kumar et al., 2020; Nakamura et al., 2020; Tan and Finkel, 2022; Xu et al., 2022). Here, we tested the hypothesis that CASM may provide a mechanistic explanation for this lipidation, undertaking a comprehensive analysis of ATG8 lipidation in the early phases of lysosome damage. Using a panel of complementary biochemical, pharmacological, and genetic approaches, we show that the robust ATG8 lipidation observed at lysosomes rapidly after damage indeed occurs via the CASM pathway. Furthermore, we provide evidence that lysosome damage-induced CASM engages the lipid transfer protein ATG2, building further links between this protein and lysosome homeostasis. Finally, we discover that ATG2 engagement represents a shared downstream response to a range of CASM stimuli, suggesting this pathway may have a broader role in the response to stress at various membranes.

## Results and discussion

### Rapid ATG8 lipidation in response to lysosome damage is largely independent of canonical autophagy

To visualize early ATG8 dynamics during lysosome damage, we used live-cell imaging to monitor GFP-LC3A in MCF10A cells. Addition of lysosome damaging agents, *L*-leucyl-leucine methyl ester (LLOMe) or glycyl-l-phenylalanine 2-naphthylamide (GPN), induced rapid and robust relocalization of GFP-LC3A, in <12 min (Fig. 1 A; Videos 1 and 2). Consistent with this, GFP-LC3A can be detected by Western blotting upon LLOMe treatment, with the appearance of the characteristic, band-shifted LC3-II form (Fig. 1 B). To investigate the mechanism underlying early ATG8 lipidation upon lysosome damage, we used genetic and pharmacological approaches to inhibit canonical autophagy. ATG13KO cells, which are deficient for canonical autophagy, or wild-type (WT) controls were treated with LLOMe to promote lysosome damage or with PP242 as a control to induce canonical autophagy. As expected, only WT cells responded to PP242 (Fig. 1, C and D). However, both WT and ATG13 KO cells displayed robust GFP-LC3A relocalization in response to LLOMe, suggesting this response occurs independently of canonical autophagy. Consistent with this, inhibition of canonical autophagy, using a Vps34 inhibitor (IN-1), or chelation of intracellular calcium (BAPTA-AM), blocked the GFP-LC3A response to PP242 but had no effect on LLOMe-induced GFP-LC3A relocalization (Fig. 1, C and D), further indicating independence from canonical autophagy. We then expanded the analysis to all GFP-tagged ATG8 family members and found that all isoforms relocalized in ATG13 KO cells upon LLOMe treatment (Fig. S1 A). Finally, as a complementary approach, we assessed the dynamics of an early autophagosome marker, WIPI2. As expected, PP242 stimulation induced WIPI2 puncta in WT cells but not ATG13 KO. Similarly, LLOMe induced a small increase in WIPI2 puncta in WT cells but also had no effect in ATG13 KO cells (Fig. 1, E and F), further confirming that the LC3A response to LLOMe occurs largely in the absence of any autophagosome formation at this time point.

Together, these data clearly demonstrate that early ATG8 lipidation induced by lysosome damage is largely independent from canonical autophagy; these findings align with observations from other recent work (Jia et al., 2022; Kumar et al., 2020; Nakamura et al., 2020; Tan and Finkel, 2022; Xu et al., 2022). While conventional lysophagy occurs, this represents only a minor portion of the observed ATG8 lipidation at early timepoints, suggesting that multiple autophagy-related pathways drive temporally and mechanistically distinct ATG8 responses to lysosome damage.

### ATG8s are recruited directly to damaged lysosomes and are associated with their tubulation

While GFP-LC3A did not colocalize with WIPI2 during early lysosome damage, we observed strong colocalization with the lysosome marker, LAMP1, in response to LLOMe (Fig. 2 A). The degree of colocalization under these conditions was even greater than that detected during canonical autophagy induction, where LC3 is expected to localize to both LAMP1-negative autophagosomes and LAMP1-positive autophagolysosomes (Fig. 2 B). We

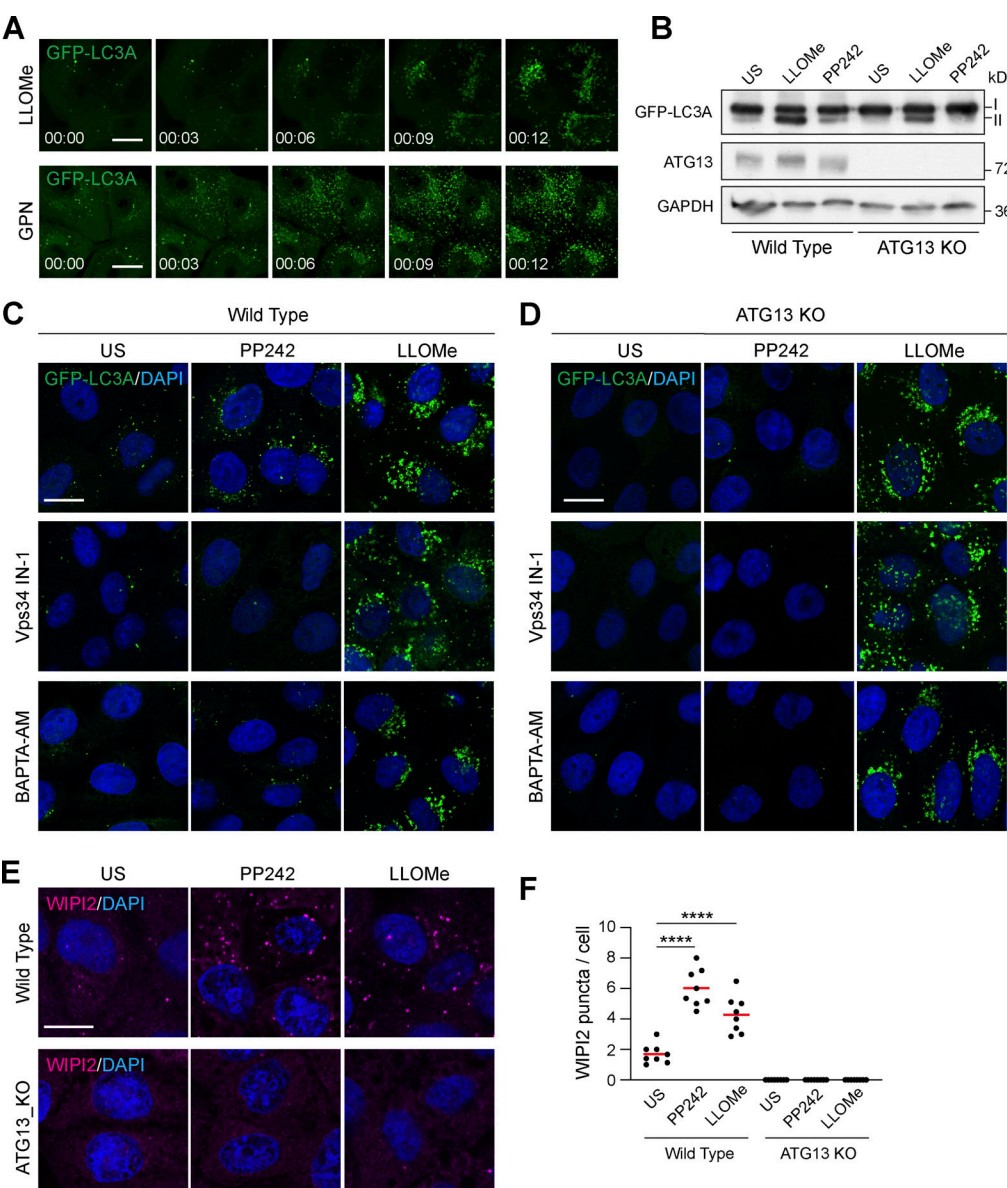

Figure 1. **Lysosome damage induces LC3A lipidation independent of canonical autophagy. (A)** Representative time-lapse confocal z stack images of MCF10A cells expressing GFP-LC3A treated with LLOMe (250 µM) or GPN (200 µM). Scale bar, 10 µm; h:min. **(B)** WT and ATG13 KO MCF10A cells expressing GFP-LC3A were treated with LLOMe (250 µM, 20 min) or PP242 (1 µM, 1 h). Western blotting was performed to probe for GFP-LC3A (I and II forms are marked). **(C and D)** Representative confocal images of GFP-LC3A in (C) WT and (D) ATG13 KO MCF10A cells treated with LLOMe (250 µM, 20 min) or PP242 (1 µM, 1 h), after pretreatment with Vps34 IN-1 (5 µM) or BAPTA-AM (10 µM). Scale bar, 10 µm. **(E)** Representative confocal images of WT and ATG13 KO MCF10A cells treated as in C and D, and stained for WIPI2. Scale bar, 10 µm. **(F)** Quantification of WIPI2 puncta from E. >100 cells were analyzed from eight fields of view in two experiments. ****, P < 0.0001, unpaired t test. Source data are available for this figure: SourceData F1.

then performed LysoIP enrichment of lysosomes (Abu-Remaileh et al., 2017) in ATG13 KO cells and observed lipidated endogenous LC3B in the lysosome fraction after LLOMe treatment (Fig. 2 C). These data suggest that LC3 is directly recruited to the lysosome membrane upon LLOMe treatment.

To examine this process further, live-cell imaging studies were performed, tracking GFP-LC3A localization shortly after lysosome damage. These experiments uncovered a striking and highly dynamic pattern of GFP-LC3A localization (Fig. 2, D and E; Videos 3 and 4), not well preserved in fixed samples. We observed the formation of highly dynamic, GFP-LC3A positive

tubules, which appear to emanate from GFP-LC3A puncta. Similar tubule formation was observed with all other ATG8 family members (Fig. S1 B). Using a LAMP1-RFP reporter, we found that while the GFP-LC3A-positive puncta are also positive for LAMP1, the distinctive GFP-LC3A tubules are LAMP1 negative (Fig. 2 F). Strikingly, the tubules frequently undergo scission events, resulting in the formation of smaller GFP-LC3A positive vesicles (Fig. 2 G; Videos 5 and 6). Tubule formation and movement are blocked by nocodazole, suggesting a requirement for microtubules (Fig. 2 H). The tubules still formed in ATG13 KO cells and did not increase with Vps34 inhibition or

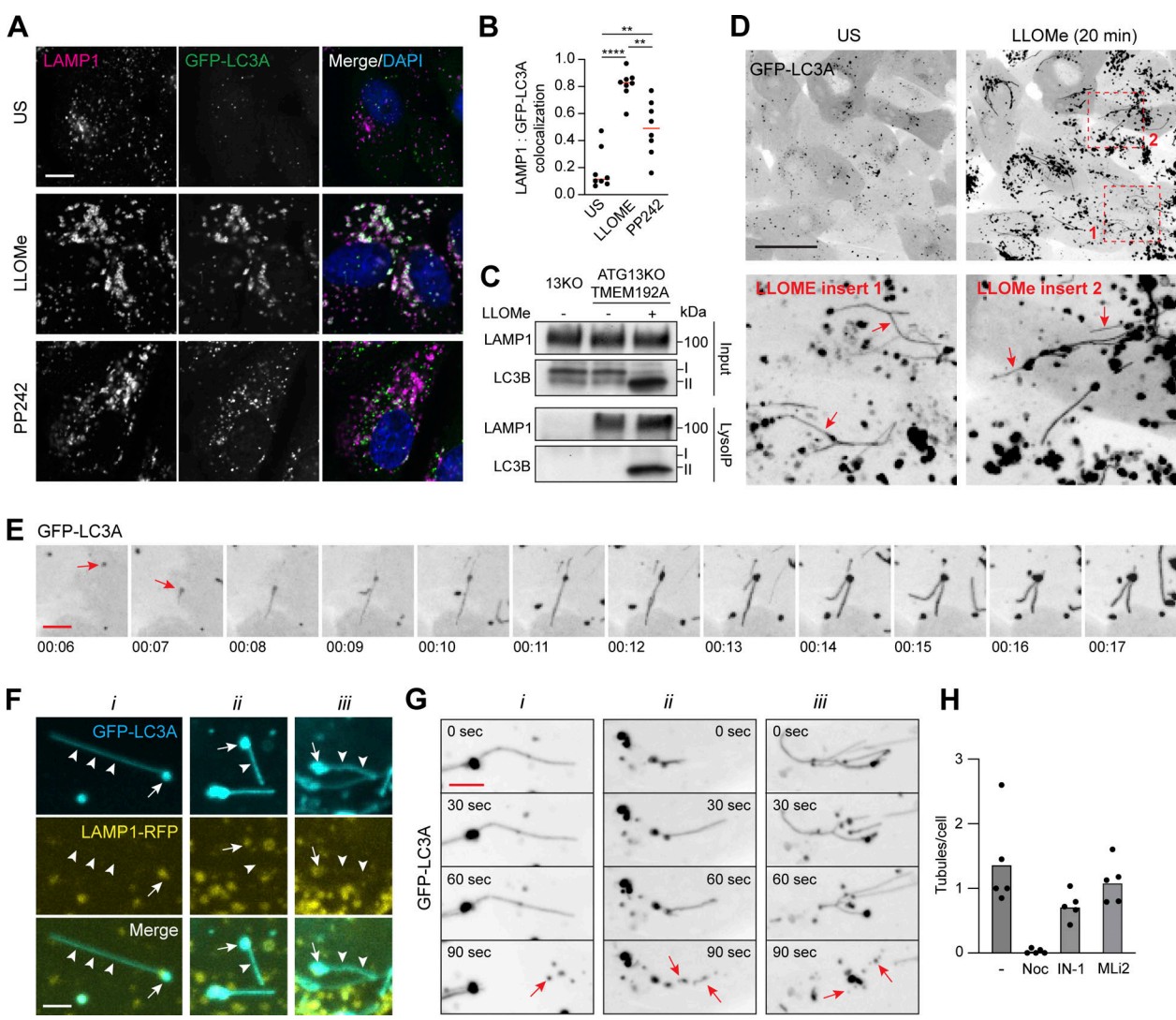

Figure 2. **LLOMe induces LC3 recruitment directly to lysosomes and their tubulation. (A)** Representative confocal images of MCF10A cells expressing GFP-LC3A, treated with LLOMe (250 µM, 20 min) or PP242 (1 µM, 1 h) and stained for LAMP1. Colocalization analysis is shown. Scale bar, 5 µm. **(B)** Quantification of GFP-LC3A overlap with LAMP1. Eight fields of view from two experiments were analyzed; ****, P < 0.0001, **, P < 0.004, unpaired t test. **(C)** Western blot analysis of input and LysoIP fractions from control or ATG13 KO cells expressing TMEM192-3xHA treated with LLOMe (250 µM, 20 min). **(D)** Images from time-lapse confocal microscopy before and after 20 min LLOMe (250 µM) treatment. Inserts show two areas of LLOMe treated cells. Arrows denote GFP-LC3A positive tubules. Scale bar, 30 µm. **(E)** Images from time-lapse series showing the formation of GFP-LC3A puncta following LLOMe (250 µM) treatment (red arrow) and subsequent tubulation. min:s. Scale bar, 3 µm. **(F)** MCF10A cells coexpressing GFP-LC3A and LAMP1-RFP treated with LLOMe (250 µM). Images show three examples (i, ii, iii). Arrows denote puncta and arrowheads denote tubules. Scale bar, 2 µm. **(G)** Time-lapse confocal images of LLOMe-induced GFP-LC3A tubulation. Three examples are shown (i, ii, iii), red arrows denote the vesiculation of tubules. Scale bar, 2 µm. **(H)** Quantification of tubules in cells treated with LLOMe (250 µM). Number of tubules per cell were analyzed in five fields of view from cells pretreated with nocodazole (1 µM), Vps34 IN-1 (5 µM), or LRRK2 inhibitor MLi2 (1 µM). A total of >150 cells per condition were analyzed. Source data are available for this figure: SourceData F2.

decrease with LRRK2 inhibition, so they seem unlikely to be related to autophagic lysosome reformation (ALR; Munson et al., 2015) or lysosome tubulation/sorting driven by LRRK2 (LYTL; Fig. 2 H and Fig. S2 C; Bonet-Ponce et al., 2020). A recent report has described LLOMe-induced lysosome tubulation involving LC3 interaction with TBC1D15 (Bhattacharya et al., 2023). However, this process appeared to require canonical autophagy and occurred at later times following damage.

Together, these data indicate that during the early phases of lysosome damage, ATG8s are lipidated directly onto lysosomes

from where they can be incorporated into highly dynamic tubular structures.

## Lysosome damage–induced ATG8 conjugation bears the hallmarks of CASM

We next considered the possible molecular mechanisms underlying the rapid and direct recruitment of ATG8 to lysosomes upon damage. Given the process is independent of canonical autophagy and involves targeting single-membrane endolysosomes, we reasoned that CASM represented a strong candidate

mechanism and tested this possibility using a series of bio-chemical, pharmacological, and genetic approaches.

First, we assessed the molecular profile of ATG8 lipidation associated with lysosome damage. CASM, unlike canonical autophagy, drives the alternative conjugation of ATG8 to phosphatidylserine (PS), providing a diagnostic molecular feature to distinguish between these related autophagy pathways (Durgan et al., 2021). To characterize ATG8 lipidation following lysosome damage, mass spectrometric analysis of GFP-LC3A was undertaken in WT cells –/+ LLOMe treatment. Importantly, LLOMe induced the formation of GFP-LC3A-PS, as well as GFP-LC3A-PE (Fig. 3, A and B), consistent with the signature of CASM. V-ATPase is a critical player in CASM and inhibition of V-ATPase with Bafilomycin A1 (BafA1) characteristically blocks this pathway (Florey et al., 2015; Jacquin et al., 2017), while simultaneously elevating levels of canonical autophagosomes. We found that BafA1 reduced the formation of both GFP-LC3A-PS and GFP-LC3A-PE following LLOMe treatment, again consistent with the hallmark features of CASM.

To interrogate this mechanism further, we combined pharmacological and genetic approaches to disrupt CASM or canonical autophagy for imaging-based analyses. In WT MCF10A cells, BafA1 dramatically reduced the GFP-LC3A response to LLOMe, consistent with CASM, while having the expected opposing effect of increasing PP242-induced GFP-LC3A puncta by inhibiting canonical autophagic flux (Fig. 3 C). Moreover, in ATG13 KO cells, which lack basal autophagosomes, BafA1 completely blocked the GFP-LC3A response to LLOMe (Fig. 3 C). In agreement with these findings, we found that LLOMe similarly induces endogenous LC3 puncta in a manner blocked by BafA1 but unaffected by Vps34 inhibition (Fig. 3 D). Importantly, the inhibitory effects of BafA1 cannot be attributed to the suppression of lysosome damage as galectin 3 puncta was still formed (Fig. 3 E). Interestingly, in WT cells, BafA1 reduced the colocalization of GFP-LC3A with Galectin 3, suggesting that, in these cells, the residual GFP-LC3A puncta represent background autophagosomes (Fig. 3, E and F).

In a final, complementary approach to exclude the input of canonical autophagy, we analyzed a more precisely defined endolysosomal compartment, completely unrelated to autophagosomes. Using latex bead engulfment to induce phagocytosis, we observed LLOMe-induced recruitment of GFP-LC3A to LAMP1-positive, latex bead–containing phagosomes (Fig. S2, A and B), again in a BafA1 sensitive manner.

Together, these data confirm that LLOMe-induced ATG8 lipidation bears several hallmark features of CASM, including the molecular signature of ATG8-PS and the distinctive pharmacological profile of BafA1 sensitivity and Vps34 insensitivity.

### Lysosome damage harnesses the V-ATPase-ATG16L1 axis

A shared feature of CASM stimuli is the disruption of endolysosomal ion and pH balances, which in turn drive increased engagement of V-ATPase V0-V1 sectors (Hooper et al., 2022). We reasoned that if lysosome damage drives ATG8 lipidation via CASM, it would also be expected to increase V0-V1 engagement. Using membrane fractionation and Western blotting, translocation of the cytosolic V1A subunit to membranes was measured

as a read-out. Strikingly, LLOMe treatment, similar to known inducers of CASM such as monensin and SaliPhe, increased translocation of the cytosolic V1A subunit to membranes in a BafA1-sensitive manner (Fig. 3 G).

The next step in CASM involves the V-ATPase-ATG16L1 axis through which increased V0-V1 association promotes the direct recruitment of ATG16L1 via its C-terminal WD40 domain, dependent upon the K490 residue (Hooper et al., 2022; Ulferts et al., 2021). An ATG16L1 K490A mutant is deficient for CASM, but competent for canonical autophagy, providing a specific genetic tool with which to dissect these pathways. Using a panel of cells, including WT, ATG16L1 KO, and KOs re-expressing either WT or K490A ATG16L1, we analyzed the molecular response to lysosome damage. ATG16L1 KO cells do not undergo GFP-LC3 lipidation upon LLOMe treatment, as detected by Western blotting, and this feature can be restored by expression of WT ATG16L1 but not the CASM deficient K490A mutant (Fig. 3 H). Similar K490 dependency was observed in the GFP-LC3A response to LLOMe using immunofluorescence confocal imaging (Fig. 3 I). Importantly, we also found that when lysosome damage is induced by monosodium urate (MSU) crystals, a more physiological stimulus, recruitment of GFP-LC3A to LAMP1-positive MSU-containing compartments, was similarly dependent on ATG16L1 K490 (Fig. 3 I).

Finally, the interaction between V-ATPase and ATG16L1 during CASM is inhibited by the *Salmonella* effector protein, SopF (Xu et al., 2019), which acts in concert with ARF1 to ribosylate the V0c subunit of V-ATPase. Strikingly, we found that expression of mCherry-SopF significantly impaired LLOMe-induced GFP-LC3A puncta, as judged by fluorescent microscopy (Fig. 3, J and K).

Together, these findings demonstrate comprehensively that lysosome damage, triggered by multiple stimuli, potently activates CASM via the V-ATPase-ATG16L1 axis as the primary early ATG8 response.

### Lipidated LC3s interact with ATG2 proteins during CASM

ATG8 proteins perform a range of important functions in autophagy-related processes, often dependent on interaction with proteins that harbor LC3 interacting regions (LIR). In canonical autophagy, LIR-based binding is critical for both cargo recognition and maturation of autophagosomes (Johansen and Lamark, 2020). Surprisingly little is known about the downstream molecular mechanisms of CASM, and a key open question is whether ATG8 proteins similarly recruit effector proteins to endolysosomal membranes.

Mammalian ATG2A and ATG2B are lipid transfer proteins that interact with ATG8s in a LIR-dependent manner (Bozic et al., 2020). They play a critical role in supplying lipids, such as PS and PE, to support autophagosome expansion (Osawa et al., 2019; Valverde et al., 2019) during canonical autophagy. ATG2s have recently been implicated in lysosome damage repair through the PITT pathway (Tan and Finkel, 2022). In light of these observations, and the data presented so far, we questioned whether CASM may involve ATG2 engagement during lysosome damage. Using GFP-TRAP immunoprecipitation, we found that LLOMe indeed induced a robust interaction between GFP-LC3A

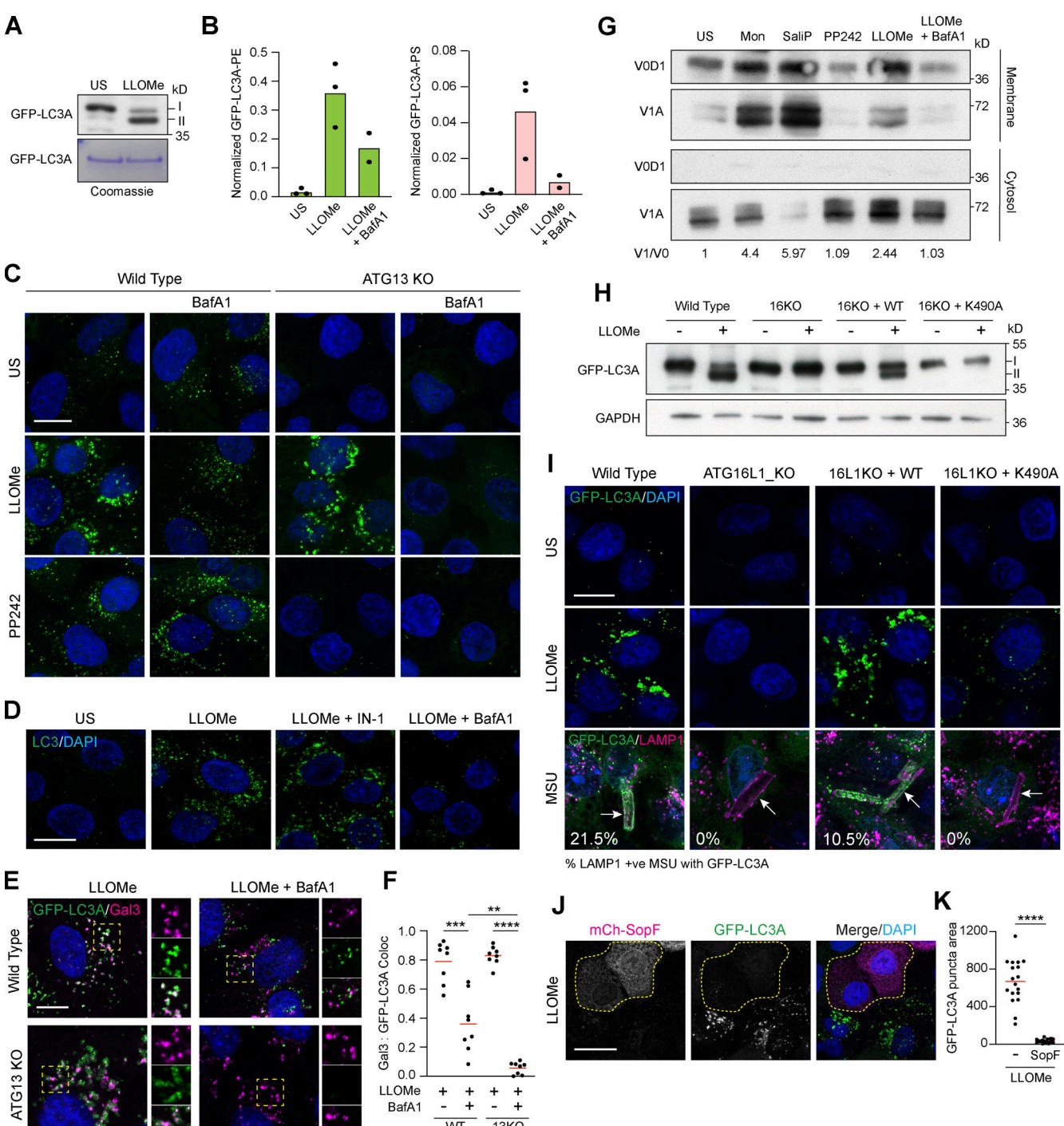

Figure 3. **Lysosome damage activates CASM through the V-ATPase-ATG16L1 axis. (A)** Western blotting and Coomassie staining of GFP-IPs from MCF10A GFP-LC3A expressing cells treated ± LLOMe (250 μM, 40 min). **(B)** Normalized mass spectrometry analysis of LC3A-PE and LC3A-PS in cells treated as in A. **(C)** Representative confocal images of GFP-LC3A in WT and ATG13 KO MCF10A cells treated with LLOMe (250 μM, 30 min) or PP242 (1 μM, 1 h), after pretreatment with BafA1 (100 nM). Scale bar, 10 μm. **(D)** Representative confocal images of endogenous LC3 in WT MCF10A cells treated as in C. Scale bar, 10 μm. **(E)** Confocal images of WT or ATG13 KO MCF10A cells expressing GFP-LC3A treated with LLOMe (250 μM, 20 min) ± BafA1 pretreatment (100 nM) and stained for Galectin-3. Inserts show single-channel cropped images. Scale bar, 5 μm. **(F)** Quantification of Gal3 overlap with GFP-LC3A from E. Eight fields of view from two experiments were analyzed; ****, P < 0.0001, ***, P < 0.0002, **, P < 0.0009, unpaired t test. **(G)** MCF10A cells were treated with monensin (100 μM, 40 min), SaliP (2.5 μM, 1 h), PP242 (1 μM, 1 h), LLOMe (250 μM, 30 min), or LLOMe + BafA1 (100 nM), 30 min. Following fractionation, membrane and cytosol fractions were probed for ATP6V1A and ATP6V0D1 by Western blotting. V1/V0 ratios are shown below. **(H)** Western blotting analysis of GFP-LC3A expressing WT MCF10A and ATG16L1 KO cells re-expressing ATG16L1 WT or K490A, treated with LLOMe (250 μM, 20 min). **(I)** Confocal images of GFP-LC3A in the MCF10A ATG16L1 cell line panel treated with LLOMe (250 μM, 20 min) or MSU crystals (200 μg/ml, 4 h) and costained for LAMP1. Arrows denote LAMP1 positive crystals, numbers show the percentage of GFP-LC3A recruitment to >100 LAMP1 positive crystals over two repeats. Scale bar, 10 μm. **(J)** Confocal images of GFP-LC3A expressing MCF10A cells cotransfected with mCherry-SopF. Outlined cells mark mCherry-SopF expressing cells. Scale bar, 20 μm. **(K)** Quantification of the area of thresholded GFP-LC3A intensity in SopF positive and negative cells from J. 18 cells from 2 experiments were analyzed; ****, P < 0.0001, unpaired t test. Source data are available for this figure: SourceData F3.

Cross et al.
Lysosome damage activates CASM

and ATG2B (Fig. 4 A). Importantly, this interaction is independent of canonical autophagy as it still occurred in ATG13 KO cells (Fig. 4 A). Furthermore, while Vps34 inhibition blocked the GFP-LC3A interaction with both ATG2A and ATG2B induced by PP242 (canonical autophagy), it did not block the LLOMe-induced interaction (Fig. 4 B). The interaction with ATG2B is dependent on GFP-LC3A lipidation, as it was lost in ATG16L1 KO cells, and more specifically on CASM, because it could be rescued by WT ATG16L1 but not K490A-expressing cells (Fig. 4 C). Furthermore, using ATG2A/B DKO cells reconstituted with WT or mutant forms of ATG2A, we demonstrate that the interaction with endogenous LC3 is dependent on the LIR domain and not the WIPI4 binding region in ATG2A (Fig. 4 D). We next tested whether CASM affected the recruitment of ATG2 to damaged lysosomes. Using LysoIP, we found both ATG2A and ATG2B associated with damaged lysosomes. However, this association was independent of LC3 lipidation (Fig. 4 E). In addition, we found that LLOMe induced an increase in ER-lysosome contacts as detected by GFP-VAPA and LAMP1-RFP imaging, which was not blocked by an ATG7 inhibitor (Fig. 4 F). Thus, while lipidated ATG8 associated with CASM engages with ATG2 during lysosome damage, this is not essential for ER–lysosome contacts.

Together, these data indicate that CASM drives an ATG8–ATG2 interaction in response to lysosome damage. This finding provides important new molecular insights into both the downstream consequences of CASM and the central role of ATG2 in lysosome homeostasis.

In considering these findings, we wondered whether the inducible engagement of the ATG2 lipid transporter may represent a broader feature of CASM, which has been conceptualized as a response to membrane stress (Kumar et al., 2021). To test this, we assessed the ability of other CASM stimuli to induce a GFP-LC3A ATG2B interaction. Strikingly, other CASM stimuli, including monensin (lysosomotropic) in MCF10A cells and LC3-associated phagocytosis (LAP) in RAW264.7 cells also induced a GFP–LC3A–ATG2B interaction (Fig. 4, G and H; Fig. S3). Together, these findings suggest that CASM drives ATG2 engagement as a general response to lysosomal perturbation.

Through this work, we have established clearly that CASM is activated in rapid response to perturbation of lysosomal permeability (Fig. 4 G). We suggest that CASM underlies the "unconventional" ATG8 lipidation observed following lysosome damage in some recent studies (Jia et al., 2022; Kumar et al., 2020; Nakamura et al., 2020; Tan and Finkel, 2022; Xu et al., 2022). In parallel, classical lysophagy acts to clear away the damaged organelles, suggesting that multiple, distinct autophagy-related pathways contribute to the maintenance of lysosome integrity. Considering that CASM and lysophagy share the common ATG8 conjugation machinery, caution is required when interpreting results from autophagy protein knockout or ATG8 lipidation experiments in this context (and others). Our current data establish comprehensively that ATG8 proteins can be conjugated directly onto lysosome membranes by CASM and can engage with an effector protein, ATG2, in an analogous manner to canonical autophagy. However, this interacting protein will not be cargo destined for degradation, but may instead exert adapter function directly at the lysosome membrane. Notably, another ATG8 protein, GABARAP, can engage with FNIP/Folliculin at lysosome membranes during TRPML1-agonist-induced CASM, acting as an adapter to promote TFEB-mediated lysosome biogenesis (Goodwin et al., 2021). We now reveal that LC3A/B engages with ATG2 proteins at damaged lysosomes. ATG2 plays an important role in lysosome repair through the recently described PITT pathway (Tan and Finkel, 2022). Although PITT occurs independently of ATG8 lipidation and autophagy, we have now identified a CASM-dependent engagement of ATG2 that may run parallel to PITT, stabilizing or reinforcing its function. As fundamental cellular processes often engage multiple, parallel pathways to safeguard proper function and integrate complex signaling information, we propose that CASM may represent one of several, cooperative mechanisms responding to lysosome damage, involving ATG8s and/or ATG2s. Furthermore, because other CASM stimuli similarly induce ATG2 engagement, we speculate that CASM may act as part of a more generalized mechanism to maintain endolysosomal homeostasis in response to stress by engaging with lipid transport machinery.

## Materials and methods

### Antibodies

Antibodies used were rabbit polyclonal anti-ATG16L1 (Cat#8090, WB 1:1,000; Cell Signalling), rabbit polyclonal anti-LC3A/B (Cat#4108, WB 1:1,000, IF 1:100; Cell Signalling), rabbit polyclonal anti-LC3B (Cat#L7543, RRID:AB_796155, WB 1:1,000; Sigma-Aldrich), rabbit monoclonal anti-ATP6V1A (Cat#ab199326, RRID:AB_2802119, WB 1:2,000; Abcam), mouse monoclonal anti-ATP6V0d1 (Cat#ab56441, WB 1:1,000; Abcam), mouse monoclonal anti-LAMP1 (Cat#555798, IF 1:100; BD Biosciences), mouse monoclonal anti LAMP1 (Cat#611042, WB, 1:500; BD Bioscience), mouse monoclonal anti-GAPDH (Cat#ab8245, RRID:AB_2107448 WB 1:1,000; Abcam), rabbit anti-ATG13 (Cat#6940, WB 1:1,000; Cell Signalling), mouse anti-GFP (Cat#1181446000, WB 1:1,000; Roche), mouse anti-WIPI2 (Cat#MCA5780GA, IF 1:100; Bio-Rad), rabbit anti ATG2B (Cat#25155-1-AP, WB 1:1,000; ProteinTech), rabbit anti-ATG2A (Cat#23226-1-AP, WB 1:1,000; ProteinTech), mouse anti-Galectin 3 (Cat#32790, IF 1:100; Scbt), rabbit anti-b tubulin (Cat#2128, WB 1:1,000; Cell Signalling), rabbit monoclonal anti-Rab10 (Cat#8127, WB 1:1,000; Cell Signalling), rabbit monoclonal anti-pT73 Rab10 (Cat#ab230261, RRID:AB_2811274, WB 1:1,000; Abcam), Alexa Fluor 488 polyclonal goat anti-rabbit IgG (Cat#A-11034, IF 1:500; Thermo Fisher Scientific), Alexa Fluor 568 polyclonal goat anti-mouse IgG (Cat#A-11004, IF 1:500; Thermo Fisher Scientific), Alexa Fluor 568 polyclonal goat anti-rabbit IgG (Cat#A-11011, RRID: AB_143157, IF 1:500; Thermo Fisher Scientific), HRP-conjugated anti-rabbit IgG (Cat#7074, WB 1:2,000; Cell Signalling), and HRP-conjugated anti-mouse IgG (Cat#7076, RRID:AB_330924, WB 1:2,000; Cell Signaling Technology).

### Reagents

Reagents and chemicals used were LLOMe (L7393; Sigma-Aldrich), GPN (SC-252858; Scbt), BafA1 (1334; Tocris), PP242 (4257; Tocris), Monensin (M5273; Sigma-Aldrich), human serum (H2918; Sigma-Aldrich), Zymosan (Z4250; Sigma-Aldrich), murine IFNγ (315-05; Peprotech), DAPI (D9542;

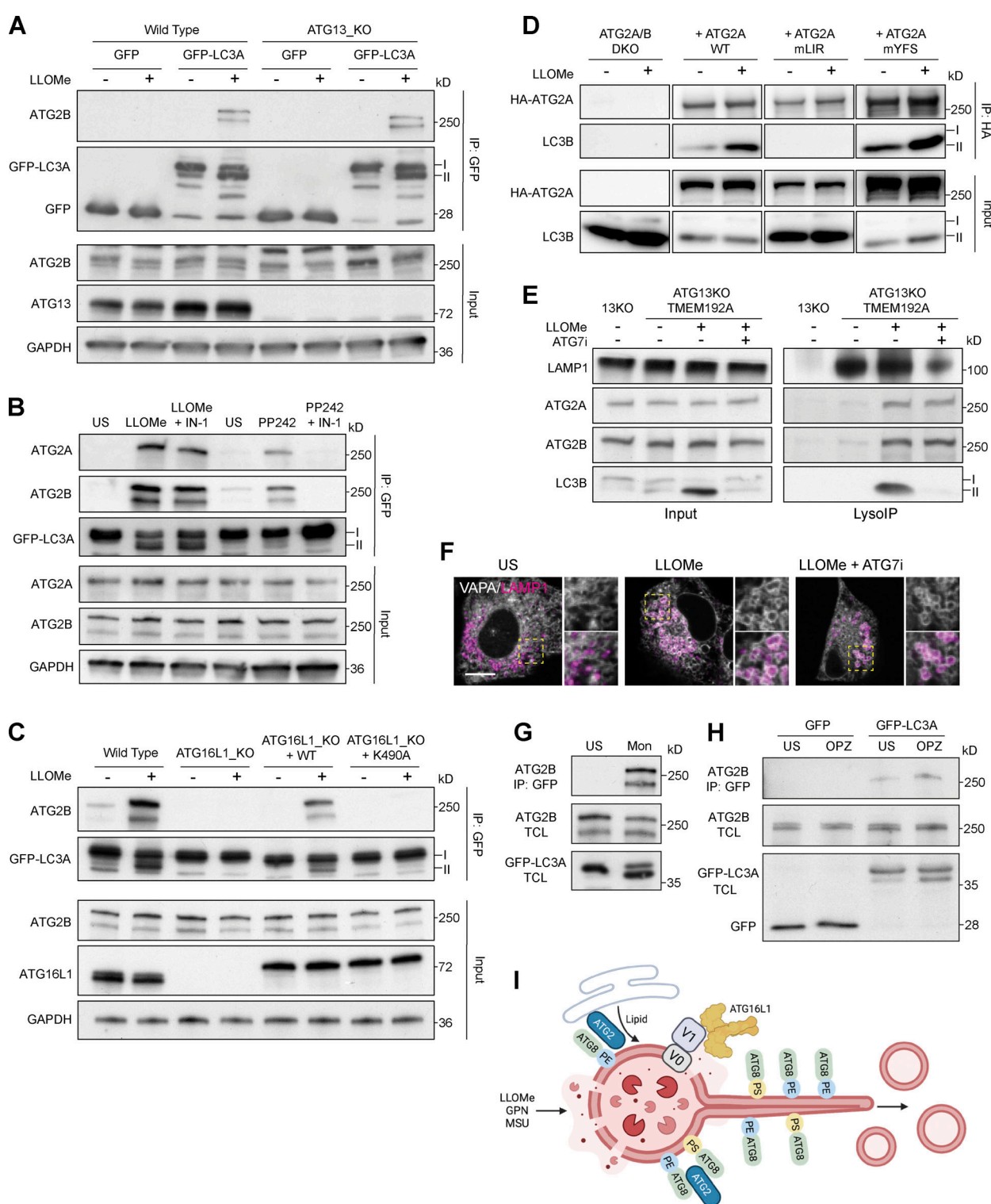

Figure 4. **LC3A interacts with ATG2B during lysosome-induced CASM. (A)** Western blot analysis of GFP-IPs and input from WT and ATG13 KO MCF10A cells expressing either GFP or GFP-LC3A-treated ± LLOMe (250 µM, 30 min). Samples were probed for ATG2B, GFP, ATG13, and GAPDH. **(B)** Western blot analysis of input and GFP-IPs from GFP-LC3A expressing MCF10A cells treated with LLOMe (250 µM, 30 min) or PP242 (1 µM, 1 h) ± Vps34 IN-1 (5 µM). Samples probed for ATG2B, ATG2A, GFP, and GAPDH. **(C)** Western blot analysis of GFP-IPs and input from MCF10A ATG16L1 cell line panel expressing GFP-LC3A treated ± LLOMe (250 µM, 30 min). Samples were probed for ATG2B, GFP, ATG16L1, and GAPDH. **(D)** Western blot analysis of input and HA-IPs from U2OS ATG2A/B DKO cells re-expressing WT, mutant LIR (mLIR), or mutant WIPI4 binding (mYFS) forms of HA-ATG2A. Cells were treated with LLOMe (250 µM, 30 min), samples were probed for ATG2A and LC3B. **(E)** Western blot analysis of input and LysoIP fractions from control or ATG13 KO cells expressing TMEM192-3xHA treated with LLOMe (250 µM, 20 min) ± ATG7 inhibitor ATG7i (10 µM). Samples probed for ATG2B, ATG2A, LAMP1, and LC3B. **(F)** Representative confocal images of ATG13 KO MCF10A cells expressing GFP-VAPA and LAMP1-RFP treated with LLOMe (250 µM, 20 min) ± ATG7 inhibitor ATG7i (10 µM). Insert shows LLOMe-induced contacts. Scale bar, 5 µm. **(G)** Western blot analysis of GFP-IPs and input from ATG13 KO MCF10A cells expressing GFP-LC3A treated with

monensin (100 µM, 40 min). Samples probed for ATG2B and GFP. **(H)** Western blot analysis of GFP-IPs and input from RAW264.7 cells expressing GFP-LC3A following incubation with opsonized zymosan (OPZ) for 40 min. Samples were probed for ATG2B and GFP. **(I)** Model of ATG8 response to lysosome damage illustrating the ATG8 associated lysosome tubulation and vesiculation, activation of the V-ATPase-ATG16L1 axis, ATG8 conjugation to PE and PS, and the engagement of lipid transfer protein ATG2. Image created using Biorender. Source data are available for this figure: SourceData F4.

Sigma-Aldrich), IN-1 (17392; Caymen Chemical), MLi2 (5756; Tocris), and saliphenylhalamide (SaliP), and TRPML1 agonist C8 and ATG7 inhibitor ATG7-IN-3 were kindly provided by Casma Therapeutics. BAPTA-AM (B1205; Thermo Fisher Scientific), nocodazole (M1404; Sigma-Aldrich), GFP-Trap (gtma-20), and control magnetic agarose beads (bmab-20) were obtained from Chromotek.

### Cell culture

WT, ATG13 KO, and ATG16L1 KO MCF10A cells (Cat#CRL-10317, RRID:CVCL_0598; ATCC) expressing GFP-LC3A (human) were prepared as described previously (Fletcher et al., 2018; Jacquin et al., 2017) and cultured in DMEM/F12 (11320074; Gibco) containing 5% horse serum (16050-122; Thermo Fisher Scientific), EGF (20 ng/mlAF-100-15; Peprotech), hydrocortisone (0.5 mg/mlH0888; Sigma-Aldrich), cholera toxin (100 ng/ml, C8052; Sigma-Aldrich), insulin (10 µg/ml, I9278; Sigma-Aldrich), and penicillin/streptomycin (100 U/ml 15140-122; Gibco) at 37°C, 5% $CO_2$.

RAW264.7 (Cat#TIB-71, RRID:CVCL_0493; ATCC) and U2OS cells (Cat#HTB-96, RRID:CVCL_0042; ATCC), described previously (Bozic et al., 2020), were maintained in DMEM (41966-029; Gibco) supplemented with 10% FBS (F9665; Sigma-Aldrich) and penicillin/streptomycin (100 U/ml, 100 µg/ml 15140-122; Gibco) at 37°C, 5% $CO_2$.

### Retrovirus generation and infection

Flag-S-ATG16L1 WT and K490A mutant in a pBabe retrovirus vector were constructed as previously described (Fletcher et al., 2018). Retrovirus was generated by transient transfection of HEK293T cells with retroviral constructs and envelope and packaging constructs using Lipofectamine 2000 (Invitrogen) following the manufacturer's guidelines. Viral supernatant was collected over 2 d. For Flag-S-ATG16L1 virus infection, MCF10A cells were seeded in a six-well plate at $5 \times 10^4$ per well. The next day, 1 ml viral supernatant was added with 10 µg/ml polybrene for 24 h followed by a media change. Cells were then selected with puromycin (2 µg/ml).

### AMAXA nucleofection

Transient transfection of GFP-VAPA and LAMP1-RFP was performed by electroporation using a Nucleofector II instrument (Lonza) and Lonza nucleofection kit V (VCA-1003; Lonza) following the manufacturer's guidelines.

### Fixed immunofluorescent confocal microscopy and analysis

Cells were seeded in 12-well plates containing coverslips and incubated at 37°C, 5% $CO_2$ for 24 h. Following treatments, cells were washed twice with ice-cold PBS and then incubated with 100% methanol at −20°C for 10 min. The cells were then washed twice with PBS and blocked with 5% BSA (A7906; Sigma-

Aldrich) in PBS for 1 h at room temperature. The cells were incubated overnight at 4°C with the primary antibodies and then washed ×3 in cold PBS. Fluorescent secondary antibodies were used at a 1:500 dilution in PBS + 5% BSA and were incubated with the cells for 1 h at room temperature. The cells were washed ×3 in cold PBS prior to being incubated with DAPI for 10 min at room temperature and then mounted onto microscope slides with ProLong Gold anti-fade reagent (P36930; Invitrogen). Image acquisition was made using a Zeiss LSM 780 confocal microscope (Carl Zeiss Ltd), using Zen software (Carl Zeiss Ltd). Image analysis and colocalization quantification was performed using ImageJ (RRID:SCR_003070).

### Live cell confocal microscopy

Cells were seeded into 35-mm glass bottom dishes (MatTek). Images stacks were acquired every 30 s using an Olympus SpinSR confocal microscope comprising Olympus IX83 stand, Olympus 60×1.5 NA UPLAPO objective lens, Yokogawa CSU-W1 scanhead, and Hamamatsu Orca Fusion camera. Images were acquired with a 2 × 2 camera bin, giving a pixel size of 108 nm. Laser excitation and emission filters for the GFP and RFP channels were 488 nm (ex) 525/50 nm (em) and 561 nm (ex) 617/73 nm (em), respectively. The image acquisition software was Olympus cellSens v4.1.

### Cell lysis, GFP-TRAP, and anti-HA immunoprecipitation

Cells expressing GFP-LC3A were seeded across multiple 15-cm dishes, treated as indicated, and then placed on ice and washed with ice-cold PBS. Each 15-cm dish was scraped into 900 µl lysis buffer (50 mM Tris pH 7.5, 150 mM NaCl, 0.5% NP40 (IGEPAL CA-630, I3021; Sigma-Aldrich), phosphatase inhibitors (1×, P0044; Sigma-Aldrich), and protease inhibitors (1×, P8340; Sigma-Aldrich). The resulting suspension was incubated on ice for 10 min and then centrifuged at 16,000 rcm, 4°C, 10 min to separate the pellet from the soluble lysate. A small fraction of the supernatant was removed for Western blotting, as described below, and the remaining lysate was subjected to preclearing and IP using magnetic beads (Chromotek) and a magnetic separation rack (Cell Signalling), according to the manufacturer's instructions. The lysate was precleared using 10 µl equilibrated magnetic agarose control beads/sample (bmab; Chromotek) for 30 min, 4°C, on a rotating wheel. Cleared lysates were then incubated with 10 µl equilibrated GFP-TRAP beads/sample (gtma; Chromotek) for 60 min, 4°C, on a rotating wheel to recover GFP-LC3A. The beads were washed 3 × for 10 min in lysis buffer at 4°C, on a rotating wheel. Enriched GFP-LC3A was eluted for analysis by Western blotting or mass spectrometry with the addition of 25 µl 2× LDS buffer (Invitrogen)/0.2 M DTT sample buffer at 100°C, 5 min.

For anti-HA immunoprecipitation, cells were washed with ice-cold PBS and scraped into lysis buffer (50 mM Tris pH 7.5,

150 mM NaCl, 0.5% NP40, IGEPAL CA-630, I3021; Sigma-Aldrich) with and protease and phosphatase cocktail inhibitors. Cleared lysates were then incubated with Anti-HA agarose slurry (A2095; Millipore) for 1 h, 4°C rotation. Samples were centrifuged at 100 × *g* and the beads were washed three times with lysis buffer.

For other samples, cells were scraped into ice-cold RIPA buffer (150 mM NaCl, 50 mM Tris–HCl, pH 7.4, 1 mM EDTA, 1% Triton X-100 (T8787; Sigma-Aldrich), 0.1% SDS (L3771; Sigma-Aldrich), 0.5% sodium deoxycholate (D6750; Sigma-Aldrich), phosphatase inhibitors (1×, P0044; Sigma-Aldrich), and protease inhibitors (1×, P8340; Sigma-Aldrich) and lysed on ice for 10 min. Lysates were centrifuged for 10 min at 10,000 *g* at 4°C.

### LysoIP
Cells stably expressing TMEM192-3xHA were seeded into 15-cm² dishes. Cells were trypsinized and resuspended in 5 ml media and treated in suspension as indicated. Cells were washed in 1 ml KPBS (136 mM KCl, 10 mM $KH_2PO_4$, pH 7.25), phosphatase inhibitors (1×, P0044; Sigma-Aldrich), and protease inhibitors (1×, P8340; Sigma-Aldrich), and resuspended in 950 µl KPBS, and 25 µl was kept for input cell lysate. Cells in the remaining sample were disrupted by vacuum pressure (Carosi et al., 2019) and transferred to prewashed magnetic anti-HA beads (88836; Thermo Fisher Scientific). Samples were rotated at 4°C for 5 min and washed × 3 in KPBS + 300 mM NaCl. Samples were eluted in 25 µl LDS sample buffer and processed for Western blotting.

### Mass spectrometric analysis of lipidated ATG8
GFP-LC3A samples were analyzed by mass spectrometry as described in Durgan et al. (2021). In brief, samples were run on 10% NuPAGE gels in MOPS buffer (Invitrogen). Each gel was washed and stained with Imperial Stain (24615; Thermo Fisher Scientific) for 2 h and then destained in d$H_2$O overnight. The gel region containing GFP-LC3A was excised into a single tube, destained, and typically saponified by treatment with 50 mM NaOH in 30% MeOH at 40°C for 2 h. The protein was digested with AspN protease (Roche) at 30°C for 16 h, in 25 mM ammonium bicarbonate. For targeted mass spectrometric assay of modified C-terminal LC3A peptides, samples were separated on a reversed-phase nanoLC column (150 × 0.075 mm; Reprosil-Pur C18AQ, Dr. Maisch), interfaced to a Q-Exactive mass spectrometer (Thermo Fisher Scientific) operating in high resolution (orbitrap) MS1 mode, with the data-dependent acquisition of low-resolution MS2 spectra generated by CID in the linear iontrap. Quantitative data were extracted using Skyline software (MacCoss Lab, University of Washington) using the sum of the chromatographic peak areas from the y1 to y10 fragment ions. Subsequently, normalization was performed against the unmodified C-terminal peptide.

### Western blotting
Western blotting was performed as described previously (Hooper et al., 2022). Briefly, cell lysates were run on SDS-PAGE gels, transferred to PVDF membrane (Immobilon-P; Millipore), blocked with 5% BSA (A7906; Sigma-Aldrich)/TBS-T for 1 h, RT, and then incubated with primary antibody at 4°C overnight. Membranes were washed 3 × for 10 min in TBS-T and incubated with HRP-conjugated secondary antibodies (7074, 7076; Cell Signalling) for 45 min, RT. Membranes were washed again 3 × for 10 min in TBS-T and then developed with ECL (RPN2209; GE). Blots were scanned (V550; Epson Perfection).

### Statistics
Statistical tests were performed using GraphPad, as indicated in figure legends.

### Online supplemental material
Fig. S1 shows the relocalization and tubulation of all GFP-tagged ATG8 isoforms in ATG13 KO MCF10A cells treated with LLOMe. Fig. S2 shows GFP-LC3A recruitment to latex bead phagosomes following LLOMe treatment, which is blocked by Bafilomycin A1 cotreatment. It also shows a positive control for the LLRK2 inhibitor. Fig. S3 shows the normalized quantification of ATG2A and ATG2B from TMT mass spectrometry analysis of GFP-LC3A pulldowns during phagocytosis.

### Data availability
The data are available from the corresponding author upon reasonable request.

## Acknowledgments
We would like to thank all members of the Florey Lab for their help. We would also like to thank the Imaging and Mass Spectrometry facilities at the Babraham Institute for their support and input.

This work was supported by grants from the Biotechnology and Biological Sciences Research Council BB/P013384/1 (BBS/E/B/000C0432 and BBS/E/B/000C0434), a Cambridge MRC DTP studentship, and D.G. McEwan and K.M. Ryan were supported by Cancer Research UK (A31287 and DRCQQR-Jun22\100001).

Author contributions: J. Cross designed, carried out, and analyzed experiments. D.G. McEwan carried out experiments and provided key reagents and advice. M. Tayler generated TMEM192 cell lines. J. Durgan designed experiments and wrote the paper. K.M. Ryan provided key resources, funding, and advice. O. Florey designed, carried out, and analyzed experiments; provided funding; and wrote the manuscript.

Disclosures: All authors have completed and submitted the ICMJE Form for Disclosure of Potential Conflicts of Interest. O. Florey reported non-financial support from Casma Therapeutics during the conduct of the study; personal fees from Casma Therapeutics outside the submitted work. No other disclosures were reported.

Submitted: 22 March 2023

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

# Supplemental material

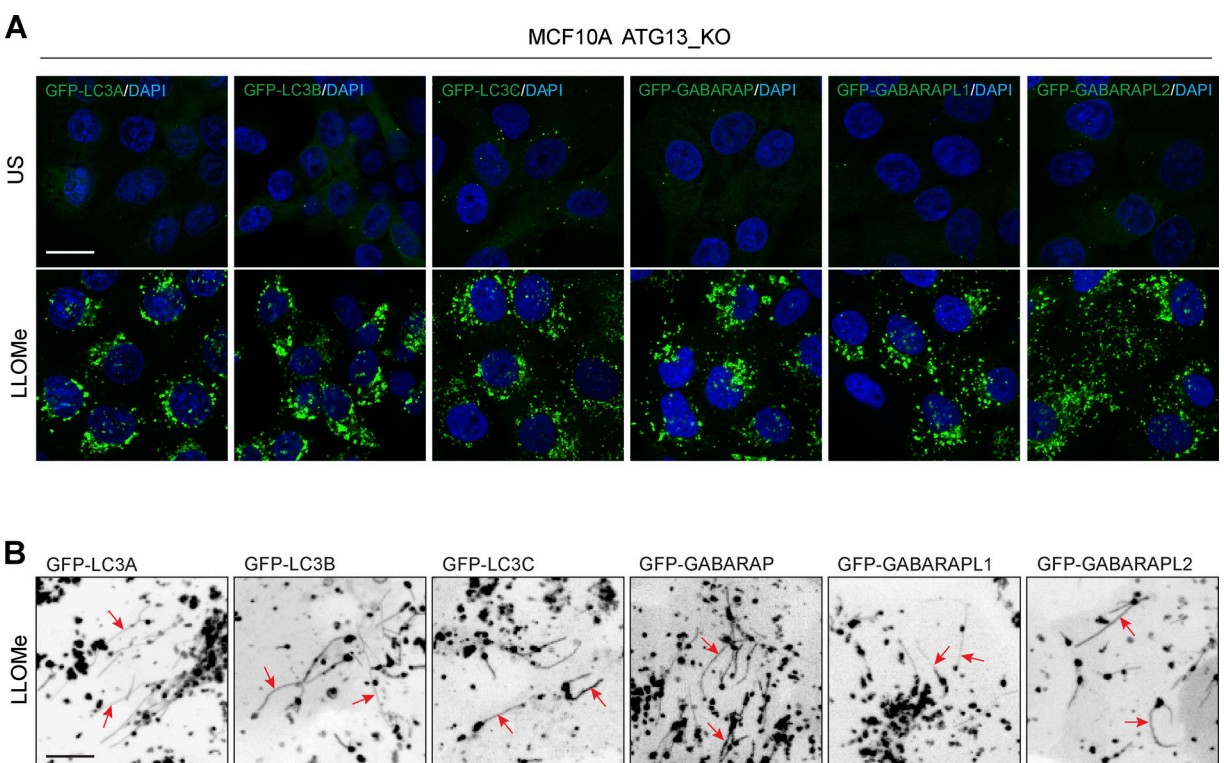

Figure S1. **Lipidation and tubule dynakics of all ATG8 family members in response to lysosome damage. (A)** Representative confocal microscopy images of ATG13 KO MCF10A cells expressing different GFP-ATG8 isoforms ± treatment with LLOMe (250 μM, 30 min). Scale bar, 10 μm. **(B)** Representative live cell confocal z-stacks in ATG13 KO MCF10A cells expressing different GFP-ATG8 isoforms treated with LLOMe (250 μM, 30 min). Arrows denote tubule formation.

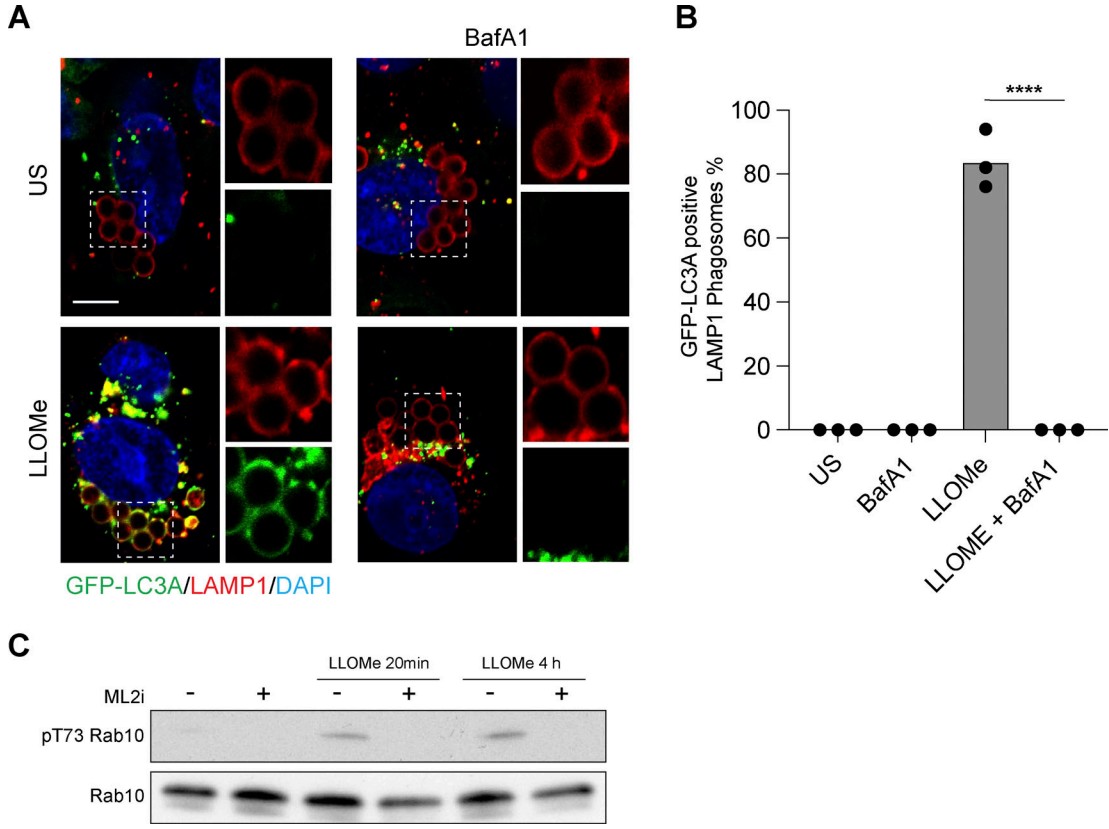

**Figure S2. Latex bead assay for CASM and validation of LRRK2 inhibitor. (A)** Representative timelapse widefield microscopy images of MCF10A cells expressing GFP-LC3A and stained for LAMP1 (red), following engulfment of 3 µm latex beads and treatment with LLOMe (250 µM, 30 min) ± BafA1 (100 nM). Cropped images show LAMP1-positive latex-bead containing phagosomes. Scale bar, 10 µm. **(B)** Quantification of GFP-LC3A recruitment to LAMP1 positive phagosomes. Data represent mean ± SEM from three independent experiments. ****, P < 0.0001, unpaired *t* test. **(C)** Western blot analysis of LRRK2 substrate, Rab10, in ATG13 KO MCF10A cells treated with LLOMe (250 µM, for 20 min or 4 h) ± LRRK2 inhibitor MLi2 (1 µM, 1.5 h pretreatment). Samples probed for Rab10 and phospho T73 Rab10. Source data are available for this figure: SourceData FS2.

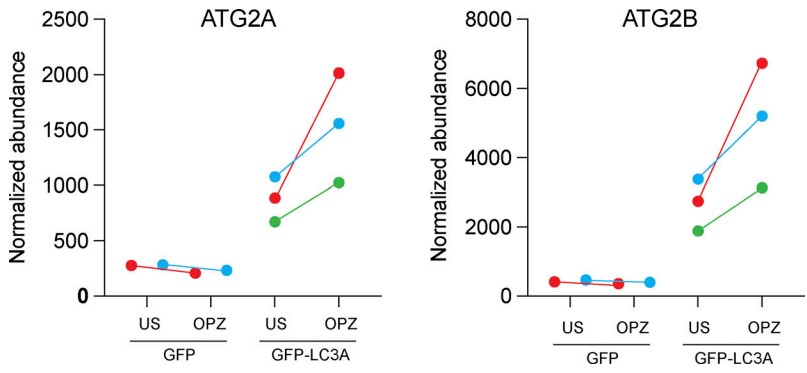

**Figure S3. Data showing mass spectrometry analysis of normalized abundance of ATG2A and ATG2B from GFP-IPs in RAW264.7 cells expressing GFP alone or GFP-LC3A treated with opsonized zymosan (OPZ) for 40 min.** Data show three independent repeats (color-coded). ATG2A/B pulldown increases in all experiments.

**Video 1. Timelapse spinning-disc confocal microscopy showing MCF10A cells expressing GFP-LC3A treated with LLOMe (250 µM).** Image stacks are acquired every 30 s. The movie plays at 4 frames/s; time min:s. Related to Fig. 1 A.

Video 2.   **Timelapse spinning-disc confocal microscopy showing MCF10A cells expressing GFP-LC3A treated with GPN (200 µM).** Image stacks are acquired every 30 s. The movie plays at 4 frames/s; time min:s. Related to Fig. 1 A.

Video 3.   **Timelapse spinning-disc confocal microscopy showing MCF10A cells expressing GFP-LC3A treated with LLOMe (250 µM).** Image stacks are acquired every 30 s. The movie plays 4 frames/s; time min:s. Related to Fig. 2 D.

Video 4.   **Cropped image sequence from MCF10A cells expressing GFP-LC3A treated with LLOMe (250 µM), showing GFP-LC3A puncta formation and tubulation.** Image stacks are acquired every 30 s. The movie plays 4 frames/s; time min:s. Related to Fig. 2 E.

Video 5.   **Cropped image sequence from MCF10A cells expressing GFP-LC3A treated with LLOMe (250 µM), showing vesiculation of GFP-LC3A tubules.** Image stacks are acquired every 30 s. The movie plays at 2 frames/s; time min:s. Related to Fig. 2 G example iii.

Video 6.   **Cropped image sequence from MCF10A cells expressing GFP-LC3A treated with LLOMe (250 µM), showing vesiculation of GFP-LC3A tubules.** Image stacks are acquired every 30 s. The movie plays 2 frames/s; time min:s. Related to Fig. 2 G example ii.

