## [Peer Review File · The Journal of Cell Biology]

Lysosome damage triggers direct ATG8 conjugation and ATG2 engagement via non-canonical autophagy

Jake Cross, Joanne Durgan, David McEwan, Matthew Tayler, Kevin Ryan, and Oliver Florey

Corresponding Author(s): Oliver Florey, Babraham Institute

Review Timeline:

Submission Date:	2023-03-22
Editorial Decision:	2023-04-16
Revision Received:	2023-07-14
Editorial Decision:	2023-08-11
Revision Received:	2023-09-01

Monitoring Editor: Hong Zhang

Scientific Editor: Tim Fessenden

Transaction Report:

DOI: <https://doi.org/10.1083/jcb.202303078>

April 16, 2023

Re: JCB manuscript #202303078

Dr. Oliver Florey
Babraham Institute
Signalling Programme The Babraham Institute
Cambridge CB22 3AT
United Kingdom

Dear Dr. Florey,

Thank you for submitting your manuscript entitled "Lysosome damage triggers direct ATG8 conjugation and ATG2 engagement via CASM". The manuscript was assessed by expert reviewers, whose comments are appended to this letter. We invite you to submit a revision if you can address the reviewers' key concerns, as outlined here.

As you will see, all reviewers expressed interest in the conceptual advance provided in this work however they agreed that certain areas needed to be strengthened prior to publication. In particular, multiple reviewers requested quantification of multiple observations and asked whether the role of LC3A described here extends to other members of the ATG8 family. In addition, a revised manuscript should resolve points 11 and 13 raised by Reviewer 3. While all reviewer comments should be addressed in some form, additional experimental data beyond those noted above are not required in a revision.

GENERAL GUIDELINES:

Text limits: Character count for a Report is < 20,000, not including spaces. Count includes title page, abstract, introduction, the joint Results & Discussion, and acknowledgments. Count does not include materials and methods, figure legends, references, tables, or supplemental legends.

Figures: Reports may have up to 5 main text figures. To avoid delays in production, figures must be prepared according to the policies outlined in our Instructions to Authors, under Data Presentation, <https://jcb.rupress.org/site/misc/ifora.xhtml>. All figures in accepted manuscripts will be screened prior to publication.

Supplemental information: There are strict limits on the allowable amount of supplemental data. Reports may have up to 3 supplemental figures. Up to 10 supplemental videos or flash animations are allowed. A summary of all supplemental material should appear at the end of the Materials and methods section.

Please note that JCB now requires authors to submit Source Data used to generate figures containing gels and Western blots with all revised manuscripts. This Source Data consists of fully uncropped and unprocessed images for each gel/blot displayed in the main and supplemental figures. Since your paper includes cropped gel and/or blot images, please be sure to provide one Source Data file for each figure that contains gels and/or blots along with your revised manuscript files. File names for Source Data figures should be alphanumeric without any spaces or special characters (i.e., SourceDataF#, where F# refers to the associated main figure number or SourceDataFS# for those associated with Supplementary figures). The lanes of the gels/blots should be labeled as they are in the associated figure, the place where cropping was applied should be marked (with a box), and molecular weight/size standards should be labeled wherever possible.

The typical timeframe for revisions is three to four months. While most universities and institutes have reopened labs and allowed researchers to begin working at nearly pre-pandemic levels, we at JCB realize that the lingering effects of the COVID-19 pandemic may still be impacting some aspects of your work, including the acquisition of equipment and reagents. Therefore, if you anticipate any difficulties in meeting this aforementioned revision time limit, please contact us and we can work with you to

find an appropriate time frame for resubmission. Please note that papers are generally considered through only one revision cycle, so any revised manuscript will likely be either accepted or rejected.

Thank you for this interesting contribution to Journal of Cell Biology. You can contact us at the journal office with any questions, cellbio@rockefeller.edu or call (212) 327-8588.

Sincerely,

Hong Zhang
Monitoring Editor
Journal of Cell Biology

Tim Fessenden
Scientific Editor
Journal of Cell Biology

Reviewer #1 (Comments to the Authors (Required)):

Cross et al report a potentially novel set of events whereby lysosomal damage triggered by several different stimuli result in LC3A lipidation and recruitment to lysosomes, which are then shown to form LAMP1 negative tubules. Such ATG8 molecules are conjugated to lipid membranes, a process this group has previously called CASM, which is independent of canonical autophagy as it proceeds in the absence of ATG13. The underlying mechanism for lysosomal damage response involves subunits of the vesicular ATPase and ATG16L1 as well as ATG16L1-dependent conjugation of LC3A to ATG2. Overall, the paper is intriguing and well described and, as such, as I have what I hope are only modest suggestions;

1. The methods mention statistical analyses, but I don't think I see any applied. Please clarify and note that two-sample t-tests are best avoided and ANOVA with post-hoc tests used if there are more than two groups.
2. In the text related to figure 2, the authors state that the observed tubules are reminiscent of LYTL. It would seem important to see if, as was reported by others, the tubules formed here can be blocked by LRRK2 kinase inhibitors such as MLI-2.
3. The paper would probably benefit from some careful proofreading. I think the figure legend to figure 3 should say V-ATPase-ATG16L1 (not L2?). There are also a few sentences where subject isn't clear ("CASM may provide a mechanistic explanation" - for?; "Consistent with this,"...with what?).

Reviewer #2 (Comments to the Authors (Required)):

In the manuscript entitled "Lysosome damage triggers direct ATG8 conjugation and ATG2 engagement via CASM", the authors reported the LC3A-associated tubular structure formation and ATG2 engagement after lysosome damage. This process is dependent on the CASM pathway, which requires V-ATPase, ATG16L1 WD40 repeats and is sensitive to SopF. Though this manuscript was submitted in a Report format, several flaws need to be reformulated or supported by data.

1. In a recent paper (PMID: 35046574), it has been shown that all ATG8 family proteins could be activated upon lysosome damage (including LLOMe treatment). And it also reported that LLOMe-induced LC3 activation relies on the V-ATPase ATG16L1 axis and is sensitive to SopF. If the CASM pathway is identical to the V-ATPase-ATG16L1 (VAIL) axis, the conclusions in Fig.1 and 3 have been proved elsewhere. Therefore, the authors should rewrite the related data.
2. Whether the LC3A-associated tubule formation is specific to LC3A? How about other ATG8 members? Is tubulin colocalized with this structure?
3. Is there a relationship between ATG2 and tubular structure? The two parts seem unrelated in the manuscript. Given that LC3A interacts with ATG2 by LLOMe, how about the localization of ATG2?
4. In the paper PMID: 32009292, it suggested that ATG2 is prone to interact with GABARAP, GABARAPL1, and only weakly with LC3A. Again, why do the authors focus only on LC3A? And the authors should also compare the interaction and localization between WT, mLIR, and mYFS ATG2 as described in the previous paper.
5. Spelling error: "ATG16L2" in Figure 3.

Reviewer #3 (Comments to the Authors (Required)):

In the paper by Cross et al. entitled "Lysosome damage triggers direct ATG8 conjugation and ATG2 engagement via CASM" the authors report on a response to lysosomal membrane damage that involves direct conjugation of Atg8/LC3 to phosphatidyl ethanolamine (PE) or phosphatidyl serine (PS) in the lysosomal membrane occurring independent of ATG13 and WIPI2 (excluding canonical autophagy), but dependent on ATG16L1 recruited to the lysosomal membrane by V-ATPase V0-V1 association. LC3 lipidated to the lysosomal membrane by ATG5-ATG12:ATG16L1 in turn recruits ATG2B as effector presumably to mediate lipid transfer from the ER to the lysosomal membrane facilitating membrane repair. LLoMe-induced recruitment of lipidated LC3A also results in lysosomal tubulation and formation of new vesicles presumably proto-lysosomes developing into primary lysosomes.

This is clearly a very interesting study adding new mechanistic insight to the CASM process applied to lysosomal membrane damage and repair. We learn that LC3 lipidated on the lysosomal membrane following damage is involved in recruitment of ATG2B to channel lipids for repair, the latter being a new effector added to the CASM pathway and adding an important missing link. This study comes in the wake of several recent studies reporting on different lysosomal membrane repair pathways and represents an important addition to better understand the processes involved in lysosomal membrane repair. The paper is timely and suitable as a Report in JCB pending required revision.

Major points and comments:

Most of the data are clear and convincing and support the conclusions made, but there is a number of figures which lack quantifications and statistics (points 4, 6-8, 10, and 12-14 below should be addressed). A second concern is that most of the data are obtained with GFP-LC3A over-expressing cells and not sufficiently backed up with data on endogenous LC3A or -B. The paper would be significantly strengthened if the authors can successfully address points 3, 11, 12 and 13 below.

1. Why is GFP-LC3A and not GFP-LC3B which is usually more highly and widely expressed (i.e. Human Protein Atlas) ?
2. Since LC3A has two isoforms differing in their N-terminal sequence which clearly can have consequences for binding specificity to partner proteins (like ATG2B looked at here) it is pertinent to ask which isoform of LC3A is used in the study?
3. Are all human ATG8s similarly recruited to the lysosomal membrane upon damage as shown here for LC3A and LC3B? Using antibodies that can distinguish LC3A and LC3B can the authors show if both are similarly recruited to the lysosome after LLoMe treatment?
The authors could do a LysolIP to show the recruitment of Atg8 proteins to the damaged lysosome to try sort out which of the endogenous Atg8s are recruited.
4. Fig 2A needs quantification of co-localization with statistics.
5. A recent paper from the group of Ivan Dikic (Nature Cell Biology <https://doi.org/10.1038/s41556-023-01125-9>) shows that after damaging lysosomes with 1 millimolar LLoMe for 2 h and wash out of LLoMe, LIR-dependent binding of TBC1D15 to Atg8s already found at the lysosomal membrane is involved in mediating tubulation and lysosome reformation via DNM2 and LIMP2 in a process similar to autophagic lysosome reformation (ALR). While in the present study tubulation is seen early, after 20 min treatment with 250 micromolar LLoMe. The mentioned paper show data that this tubulation is dependent on canonical autophagy and not CASM. Is this a tubulation response occurring much later than the one reported here by Cross et al.? Can the authors comment on this in the Discussion section to clarify this issue?
6. Colocalization between GFP-LC3A and Gal3 must be quantified with statistics in Fig. 3E.
7. The SopF experiment in Fig. 3I needs some quantification and statistics as do Fig 3H.
8. The comparative IP in Fig. 4B should be quantified and the same goes for Fig. 4C where the signal is rather weak for ATG2B in ATG16L1_KO reconstituted with WT making it hard to be certain that there is no signal for ATG2B in the IP of extracts from ATG16L1 KO cells reconstituted with the K490A mutant.
9. Can the authors explain why, despite a high expression level of reconstituted ATG16L1 WT and K490A mutant in the KO cells, is relatively little ATG2B IP compared to ATG16L1 WT cells?
10. The ATG2B blot of the membrane fraction in Fig. 4D is very blurry. This needs to be improved and quantification is required to determine if there is a statistically significant increased membrane recruitment of ATG2B upon LLoMe treatment of ATG13 KO cells relative to 16L1_KO cells.
11. The interaction data should ideally be backed up by imaging showing ER-proximal lysosomes increasing upon LLoMe of ATG16L1 KO cells reconstituted with WT but not with K490A mutant ATG16L1.
12. Both Fig. 4E and 4F should be backed up or replaced with IPs of endogenous LC3 to see possible colIP of ATG2B. The present Fig. 4F needs quantification as it is not a big difference in intensity of the ATG2B bands in the upper panel lanes 3 and 4 as observed by visual inspection.
13. Will a LIR mutant ATG2B be recruited? Reconstitute ATG2A/B DKO cells with WT and LIR mutant Atg2B to test if the LIR is required for effective recruitment of ATG2B to the LLoMe-damaged lysosomal membranes.

Minor points/corrections:

14. The WIPI dots in Fig. 1 E and F should be quantified.
15. Last sentence of the Introduction: "have" is missing to give "may have a broader role".
16. In line three from the bottom of page 10 "all" needs to be replaced with "also" to avoid overstating since only two additional CASM stimuli are tested.

17. On the top of page 11 the sentence "We suggest that CASM underlies the 'unconventional' ATG8 lipidation observed following lysosome damage in some recent studies." needs to be accompanied with references to the particular papers the authors are thinking on here.

18. Under Reagents "IN-1 (17392, Caymen)" should be corrected to IN-1 (17392, Cayman Chemical)"

We thank the reviewers for their time and diligence in evaluating our manuscript.

Reviewer #1 (Comments to the Authors (Required)):

Cross et al report a potentially novel set of events whereby lysosomal damage triggered by several different stimuli result in LC3A lipidation and recruitment to lysosomes, which are then shown to form LAMP1 negative tubules. Such ATG8 molecules are conjugated to lipid membranes, a process this group has previously called CASM, which is independent of canonical autophagy as it proceeds in the absence of ATG13. The underlying mechanism for lysosomal damage response involves subunits of the vesicular ATPase and ATG16L1 as well as ATG16L1-dependent conjugation of LC3A to ATG2. Overall, the paper is intriguing and well described and, as such, as I have what I hope are only modest suggestions;

We thank the reviewer for their consideration of our report.

1. The methods mention statistical analyses, but I don't think I see any applied. Please clarify and note that two-sample t-tests are best avoided and ANOVA with post-hoc tests used if there are more than two groups.

We have now included more statistical analysis of our data (Fig. 1F, 2B, 3F and 3K) and have now described the statistical test used in the figure legends.

2. In the text related to figure 2, the authors state that the observed tubules are reminiscent of LYTL. It would seem important to see if, as was reported by others, the tubules formed here can be blocked by LRRK2 kinase inhibitors such as MLI-2.

We have now used the LRRK2 inhibitor MLI2, which we confirm blocks LRRK2 activity using phosphor Rab10 as a readout (Fig. S2C). We find that MLI2 has no effect on LLOMe induced GFP-LC3A positive tubulation (Fig. 2H).

3. The paper would probably benefit from some careful proofreading. I think the figure legend to figure 3 should say V-ATPase-ATG16L1 (not L2?). There are also a few sentences where subject isn't clear ("CASM may provide a mechanistic explanation" - for?; "Consistent with this,"...with what?).

We thank the reviewer for picking up these errors. They have now been corrected.

Reviewer #2 (Comments to the Authors (Required)):

In the manuscript entitled "Lysosome damage triggers direct ATG8 conjugation and ATG2 engagement via CASM", the authors reported the LC3A-associated tubular structure formation and ATG2 engagement after lysosome damage. This process is dependent on the CASM pathway, which requires V-ATPase, ATG16L1 WD40 repeats and is sensitive to SopF. Though this manuscript was submitted in a Report format, several flaws need to be reformulated or supported by data.

We thank the reviewer for their careful evaluation.

1. In a recent paper (PMID: 35046574), it has been shown that all ATG8 family proteins

could be activated upon lysosome damage (including LLOMe treatment). And it also reported that LLOMe-induced LC3 activation relies on the V-ATPase ATG16L1 axis and is sensitive to SopF. If the CASM pathway is identical to the V-ATPase-ATG16L1 (VAIL) axis, the conclusions in Fig.1 and 3 have been proved elsewhere. Therefore, the authors should rewrite the related data.

We understand the point raised by the reviewer, and have cited the relevant papers. However, the papers mentioned do not explicitly define the nature of LC3 lipidation upon endomembrane damage. Indeed, one of the first reports on the V-ATPase – ATG16L1 axis, proposed it in the form of xenophagy, which is a selective autophagosome associated process. In addition, the paper (PMID: 35046574), refers to V-ATPase-ATG16L1 axis as being involved in “lysophagy”, again a selective autophagosome associated process. Thus, there is confusion within the literature, and why we feel it an important and relevant point to comprehensively characterise the ATG8 response to lysosome damage, and accurately describe the response as CASM. This both validates previous findings and extends it to the full mechanism and interpretation. We hope our study achieves this.

2. Whether the LC3A-associated tubule formation is specific to LC3A? How about other ATG8 members? Is tubulin colocalized with this structure?

We have now confirmed that all GFP-tagged isoforms of ATG8 are involved in the CASM response to lysosome damage, and are associated with tubulation (Fig. S1A and B).

3. Is there a relationship between ATG2 and tubular structure? The two parts seem unrelated in the manuscript. Given that LC3A interacts with ATG2 by LLOMe, how about the localization of ATG2?

This is an interesting point that warrants future investigation. Of note, we were still able to observe LC3 tubulation in ATG2A/B DKO cells, although we have not conducted a thorough investigation for any subtle differences. Therefore, we cannot at this stage rule out a role for the interaction in tubulation.

4. In the paper PMID: 32009292, it suggested that ATG2 is prone to interact with GABARAP, GABARAPL1, and only weakly with LC3A. Again, why do the authors focus only on LC3A? And the authors should also compare the interaction and localization between WT, mLIR, and mYFS ATG2 as described in the previous paper.

In the cited paper, authors show that there is an interaction between LC3A and ATG2 using GST pulldowns. We initially worked on LC3A due to availability of cell line reagents. We have now included extra data to show that ATG2A is able to interact with endogenous LC3, and that this interaction depends on the LIR domain of ATG2A and not the WIPI4 binding YFS domain (Fig. 4D).

5. Spelling error: "ATG16L2" in Figure 3.

We thank the reviewer for pointing this out and have corrected.

Reviewer #3 (Comments to the Authors (Required)):

In the paper by Cross et al. entitled "Lysosome damage triggers direct ATG8 conjugation and

ATG2 engagement via CASM" the authors report on a response to lysosomal membrane damage that involves direct conjugation of Atg8/LC3 to phosphatidyl ethanolamine (PE) or phosphatidyl serine (PS) in the lysosomal membrane occurring independent of ATG13 and WIPI2 (excluding canonical autophagy), but dependent on ATG16L1 recruited to the lysosomal membrane by V-ATPase V0-V1 association. LC3 lipidated to the lysosomal membrane by ATG5-ATG12:ATG16L1 in turn recruits ATG2B as effector presumably to mediate lipid transfer from the ER to the lysosomal membrane facilitating membrane repair. LLoMe-induced recruitment of lipidated LC3A also results in lysosomal tubulation and formation of new vesicles presumably proto-lysosomes developing into primary lysosomes.

This is clearly a very interesting study adding new mechanistic insight to the CASM process applied to lysosomal membrane damage and repair. We learn that LC3 lipidated on the lysosomal membrane following damage is involved in recruitment of ATG2B to channel lipids for repair, the latter being a new effector added to the CASM pathway and adding an important missing link. This study comes in the wake of several recent studies reporting on different lysosomal membrane repair pathways and represents an important addition to better understand the processes involved in lysosomal membrane repair. The paper is timely and suitable as a Report in JCB pending required revision.

We thank the reviewer for their supportive review.

Major points and comments:

Most of the data are clear and convincing and support the conclusions made, but there is a number of figures which lack quantifications and statistics (points 4, 6-8, 10, and 12-14 below should be addressed). A second concern is that most of the data are obtained with GFP-LC3A over-expressing cells and not sufficiently backed up with data on endogenous LC3A or -B. The paper would be significantly strengthened if the authors can successfully address points 3, 11, 12 and 13 below.

1. Why is GFP-LC3A and not GFP-LC3B which is usually more highly and widely expressed (i.e. Human Protein Atlas) ?

We initially focused on LC3A due to availability of cell line reagents. We have now included analysis of all LC3 and GABARAP isoforms (Fig. S1A and B).

2. Since LC3A has two isoforms differing in their N-terminal sequence which clearly can have consequences for binding specificity to partner proteins (like ATG2B looked at here) it is pertinent to ask which isoform of LC3A is used in the study?

The isoform we work on is LC3Av1 which has been confirmed with sequencing and also mass spectrometry peptide analysis.

3. Are all human ATG8s similarly recruited to the lysosomal membrane upon damage as shown here for LC3A and LC3B? Using antibodies that can distinguish LC3A and LC3B can the authors show if both are similarly recruited to the lysosome after LLoMe treatment?

The authors could do a LysoIP to show the recruitment of Atg8 proteins to the damaged lysosome to try sort out which of the endogenous Atg8s are recruited.

As mentioned, we have now included data on GFP-tagged versions of all ATG8 proteins, and find similar CASM dependent recruitment to lysosomes and association with tubulation (Fig.

S1A and B). We have also included LysoIP experiments and demonstrate recruitment of endogenous lipidated LC3 to lysosome fractions upon damage (Fig. 2C and 4E). We do not see any evidence of selective ATG8 lipidation during CASM, and instead, all isoforms can be lipidated if expressed.

4. Fig 2A needs quantification of co-localization with statistics.

We have now included quantification (Fig. 2B).

5. A recent paper from the group of Ivan Dikic (Nature Cell Biology <https://doi.org/10.1038/s41556-023-01125-9>) shows that after damaging lysosomes with 1 millimolar LLoMe for 2 h and wash out of LLoMe, LIR-dependent binding of TBC1D15 to Atg8s already found at the lysosomal membrane is involved in mediating tubulation and lysosome reformation via DNM2 and LIMP2 in a process similar to autophagic lysosome reformation (ALR). While in the present study tubulation is seen early, after 20 min treatment with 250 micromolar LLoMe. The mentioned paper show data that this tubulation is dependent on canonical autophagy and not CASM. Is this a tubulation response occurring much later than the one reported here by Cross et al.? Can the authors comment on this in the Discussion section to clarify this issue?

This paper was published after our initial submission. However, we have now included a comment within the results and discussion. As we see acute tubulation even in the absence of canonical autophagy, we feel our observations differ from those of the Dikic group. Further, as we do not see LAMP1 on our tubules, and the process is not enhanced by Vps34 inhibition, we feel it is unrelated to ALR.

6. Colocalization between GFP-LC3A and Gal3 must be quantified with statistics in Fig. 3E.

We have now included quantification (Fig. 3F).

7. The SopF experiment in Fig. 3I needs some quantification and statistics as do Fig 3H.

We have now included quantification (Fig. 3K).

8. The comparative IP in Fig. 4B should be quantified and the same goes for Fig. 4C where the signal is rather weak for ATG2B in ATG16L1_KO reconstituted with WT making it hard to be certain that there is no signal for ATG2B in the IP of extracts from ATG16L1 KO cells reconstituted with the K490A mutant.

We understand the point raise regarding ATG2 pull down in ATG16L1 KO cells reconstituted with WT. We find that the level of ATG2 pulldown correlates to the level of rescued LC3 lipidation. In the original experiments there was not a full rescue of lipidation and so only minor ATG2 interaction. We have repeated the experiment and include new blots where the level of lipidation rescue is higher, along with better ATG2 pulldown (Fig. 4C). Similarly, we have included new blots for Fig. 4B to include both ATG2A and ATG2B data.

9. Can the authors explain why, despite a high expression level of reconstituted ATG16L1 WT and K490A mutant in the KO cells, is relatively little ATG2B IP compared to ATG16L1 WT cells?

As described above. The level of ATG2 pulldown correlates to level of LC3 lipidation rescue. Even with high levels of ATG16L1 re-expression, unless there is lipidation, there will not be

ATG2 interaction. We and others have commented previously that the lipidation process can be very sensitive to ATG16L1 levels, too much or too little can potentially cause issues.

10. The ATG2B blot of the membrane fraction in Fig. 4D is very blurry. This needs to be improved and quantification is required to determine if there is a statistically significant increased membrane recruitment of ATG2B upon LLoMe treatment of ATG13 KO cells relative to 16L1_KO cells.

Thank you for raising this important issue. We agree that these experiments were not optimal, especially as they are crude membrane fractions and not specific for lysosomes. We therefore repeated the experiments using LysolP protocols, to better enrich lysosome membranes. In these new experiments (Fig. 4E), we observe a movement of both ATG2A and ATG2B to lysosome fractions of ATG13 KO cells upon lysosome damage. Importantly, this movement was not inhibited by blocking CASM/lipidation with an effective ATG7 inhibitor. Thus, we now conclude that CASM is not essential for getting ATG2 to lysosomes, but once there it engages with it, potentially enhancing its function. This is something that we intend to follow up in the future. We feel these new experiments enable us to make a much stronger conclusion and thank the reviewer for prompting us to explore further.

11. The interaction data should ideally be backed up by imaging showing ER-proximal lysosomes increasing upon LLoMe of ATG16L1 KO cells reconstituted with WT but not with K490A mutant ATG16L1.

This point relates to our new data described above. We performed new experiments in ATG13KO cells to observe ER-Lysosome contacts in live cells using GFP-VAPA as an ER marker, and RFP-LAMP1 for lysosomes. In agreement with a previous report, we found lysosome damage induced new contacts between VAPA and LAMP1, with the VAPA signal making “rings” around lysosomes. This was not altered by blocking ATG8 lipidation (Fig. 4F). Thus, in agreement with new data in Fig. 4 E, our model is that lysosome damage promotes ER-lysosome contacts (likely through the PITT pathway), and that during this, CASM will interact and engage with ATG2. We have altered the text to reflect this.

12. Both Fig. 4E and 4F should be backed up or replaced with IPs of endogenous LC3 to see possible coIP of ATG2B. The present Fig. 4F needs quantification as it is not a big difference in intensity of the ATG2B bands in the upper panel lanes 3 and 4 as observed by visual inspection.

We have now performed and included data of ATG2 IP and show pulldown of endogenous LC3 (Fig. 4D). With regards to the figure showing LC3-ATG2B interaction during phagocytosis, we now include supplementary data from a quantitative mass spectrometry analysis of GFP-LC3A interactors during phagocytosis of zymosan. In all 3 repeats we see an increase pulldown of both ATG2A and ATG2B (Fig. S3).

13. Will a LIR mutant ATG2B be recruited? Reconstitute ATG2A/B DKO cells with WT and LIR mutant Atg2B to test if the LIR is required for effective recruitment of ATG2B to the LLoMe-damaged lysosomal membranes.

We have now included new data to confirm that the LIR domain of ATG2A, and not its WIPI4 binding YFS domain, is required for its interaction with LC3 during LLOMe induced CASM (Fig. 4D).

Minor points/corrections:

14. The WIPI dots in Fig. 1 E and F should be quantified.

We have now included quantification (Fig. 1F).

15. Last sentence of the Introduction: "have" is missing to give "may have a broader role".

We thank the reviewer for pointing this out and have corrected.

16. In line three from the bottom of page 10 "all" needs to be replaced with "also" to avoid overstating since only two additional CASM stimuli are tested.

We thank the reviewer for pointing this out and have corrected.

17. On the top of page 11 the sentence "We suggest that CASM underlies the 'unconventional' ATG8 lipidation observed following lysosome damage in some recent studies." needs to be accompanied with references to the particular papers the authors are thinking on here.

We thank the reviewer for pointing this out and have corrected.

18. Under Reagents "IN-1 (17392, Caymen)" should be corrected to IN-1 (17392, Cayman Chemical)"

We thank the reviewer for pointing this out and have corrected.

August 11, 2023

RE: JCB Manuscript #202303078R

Dr. Oliver Florey
Babraham Institute
Signalling Programme The Babraham Institute
Cambridge CB22 3AT
United Kingdom

Dear Dr. Florey:

Thank you for submitting your revised manuscript entitled "Lysosome damage triggers direct ATG8 conjugation and ATG2 engagement via CASM". We would be happy to publish your paper in JCB pending final revisions necessary to meet our formatting guidelines (see details below) as well as minor text edits requested by Reviewer 3.

Your manuscript was handled as a co-submission with MS 202302067 by Kuwahara and colleagues, however we have yet to receive a revision of that work. The decision to await their work for co-publication is yours. Please alert us if indeed you would like to wait or if you have any questions.

A. MANUSCRIPT ORGANIZATION AND FORMATTING:

Full guidelines are available on our Instructions for Authors page, <http://jcb.rupress.org/submission-guidelines#revised>. Submission of a paper that does not conform to JCB guidelines will delay the acceptance of your manuscript.

1) Text limits: Character count for Reports is < 20,000, not including spaces. Count includes abstract, introduction, results, discussion, and acknowledgments. Count does not include title page, figure legends, materials and methods, references, tables, or supplemental legends.

2) Figures limits: Reports may have up to five main figures and three supplemental figures/tables.

3) Figure formatting: Scale bars must be present on all microscopy images, including inset magnifications. Please avoid pairing red and green for images and graphs to ensure legibility for color-blind readers. If red and green are paired for images, please ensure that the particular red and green hues used in micrographs are distinctive with any of the colorblind types. If not, please modify colors accordingly or provide separate images of the individual channels. Molecular weight or nucleic acid size markers must be included on all gel electrophoresis.

** Please add scale bars to images in Figure 2 E-G.

** Please include molecular weight markers on all Western blots.

4) Statistical analysis: Error bars on graphic representations of numerical data must be clearly described in the figure legend. The number of independent data points (n) represented in a graph must be indicated in the legend. Statistical methods should be explained in full in the materials and methods. For figures presenting pooled data the statistical measure should be defined in the figure legends. Please also be sure to indicate the statistical tests used in each of your experiments (either in the figure legend itself or in a separate methods section) as well as the parameters of the test (for example, if you ran a t-test, please indicate if it was one- or two-sided, etc.). Also, if you used parametric tests, please indicate if the data distribution was tested for normality (and if so, how). If not, you must state something to the effect that "Data distribution was assumed to be normal but this was not formally tested."

5) Abstract and title: The abstract should be no longer than 160 words and should communicate the significance of the paper for a general audience. The title should be less than 100 characters including spaces. Make the title concise but accessible to a general readership.

** The current title includes the abbreviation "CASM" with which readers would not be familiar. Please modify the title to omit this abbreviation as you see fit. An alternative could be "Direct conjugation of ATG8 mediates repair of damaged lysosomes independent of lysophagy."

6) Materials and methods: Should be comprehensive and not simply reference a previous publication for details on how an experiment was performed. Please provide full descriptions in the text for readers who may not have access to referenced manuscripts.

** Please provide details on the generation of retrovirus.

7) Please be sure to provide the sequences for all of your primers/oligos and RNAi constructs in the materials and methods. You must also indicate in the methods the source, species, and catalog numbers (where appropriate) for all of your antibodies. Please also indicate the acquisition and quantification methods for immunoblotting/western blots.

8) Microscope image acquisition: The following information must be provided about the acquisition and processing of images:

- a. Make and model of microscope
- b. Type, magnification, and numerical aperture of the objective lenses
- c. Temperature
- d. Imaging medium
- e. Fluorochromes
- f. Camera make and model
- g. Acquisition software
- h. Any software used for image processing subsequent to data acquisition. Please include details and types of operations involved (e.g., type of deconvolution, 3D reconstitutions, surface or volume rendering, gamma adjustments, etc.).

10) Supplemental materials: There are strict limits on the allowable amount of supplemental data. Reports may have up to 3 supplemental figures. Please also note that tables, like figures, should be provided as individual, editable files. A summary of all supplemental material should appear at the end of the Materials and methods section.

13) ORCID IDs: ORCID IDs are unique identifiers allowing researchers to create a record of their various scholarly contributions in a single place. Please note that ORCID IDs are now *required* for all authors. At resubmission of your final files, please be sure to provide your ORCID ID and those of all co-authors.

Please note that JCB requires authors to submit Source Data used to generate figures containing gels and Western blots with all revised manuscripts. This Source Data consists of fully uncropped and unprocessed images for each gel/blot displayed in the main and supplemental figures. File names for Source Data figures should be alphanumeric without any spaces or special characters (i.e., SourceDataF#, where F# refers to the associated main figure number or SourceDataFS# for those associated with Supplementary figures). The lanes of the gels/blots should be labeled as they are in the associated figure, the place where cropping was applied should be marked (with a box), and molecular weight/size standards should be labeled wherever possible. Source Data files will be made available to reviewers during evaluation of revised manuscripts and, if your paper is eventually published in JCB, the files will be directly linked to specific figures in the published article.

Journal of Cell Biology now requires a data availability statement for all research article submissions. These statements will be published in the article directly above the Acknowledgments. The statement should address all data underlying the research presented in the manuscript. Please visit the JCB instructions for authors for guidelines and examples of statements at (<https://rupress.org/jcb/pages/editorial-policies#data-availability-statement>).

B. FINAL FILES:

Thank you for this interesting contribution, we look forward to publishing your paper in Journal of Cell Biology.

Sincerely,

Hong Zhang
Monitoring Editor
Journal of Cell Biology

Tim Fessenden
Scientific Editor
Journal of Cell Biology

Reviewer #2 (Comments to the Authors (Required)):

The revised manuscript answers most of the questions. Although this version does not fully answer the biological relevance of tubular structures and ATG2 engagement, it still provides some insights into this new field. I recommend publication in JCB Journal in report format.

Reviewer #3 (Comments to the Authors (Required)):

The authors have addressed all my comments in a more than satisfactory manner.

Line 158: At first mention of the LysolP procedure a reference to PMID: 29074583 should be inserted.

Line 177: Delete "I"

Line 284: Correct "lapidated" to "lipidated" (Be aware that your Mac can not be trusted as it will auto correct lipidated to lapidated. Do not ask me why!)

Lines 340-342: In the Contributions section the added author Kevin Ryan needs to be mentioned.

We thank the reviewers and editors for their time and diligence in evaluating our manuscript.

Reviewer #3

Line 158: At first mention of the LysolP procedure a reference to PMID: 29074583 should be inserted.

Line 177: Delete "I"

Line 284: Correct "lapidated" to "lipidated" (Be aware that your Mac can not be trusted as it will auto correct lipidated to lapidated. Do not ask me why!)

Lines 340-342: In the Contributions section the added author Kevin Ryan needs to be mentioned.

We thank the reviewer and have made corrections for these points.

Formatting comments

** Please add scale bars to images in Figure 2 E-G.

** Please include molecular weight markers on all Western blots.

We have made these changes to figures.

** The current title includes the abbreviation "CASM" with which readers would not be familiar. Please modify the title to omit this abbreviation as you see fit.

We have changed the title to *"Lysosome damage triggers direct ATG8 conjugation and ATG2 engagement via non-canonical autophagy"*

** Please provide details on the generation of retrovirus.

This has been included in the methods section.